



# Influence of adsorption of CO₂ on cylinder and fractionation of CO₂ and air during preparation of a standard mixture

Nobuyuki Aoki[1], Shigeyuki Ishidoya[2], Shohei Murayama[2], and Nobuhiro Matsumoto[1]

[1]National Metrology Institute of Japan, National Institute of Advanced Industrial Science and Technology (NMIJ/AIST), 1-1-1 Umezono, Tsukuba 305-8563, Japan

[2]Environmental Management Research Institute, National Institute of Advanced Industrial Science and Technology (EMRI/AIST), Tsukuba 305-8569, Japan

*Correspondence to*: Nobuyuki Aoki (aoki-nobu@aist.go.jp) Tel: +81-29-861-6824: fax: +81-29-861-

10    6854.

**Abstract:** We conducted a study to fully understand carbon dioxide (CO₂) adsorption to a cylinder's internal surface and fractionation of CO₂ and air during the preparation of standard mixtures of atmospheric CO₂ levels through a multistep dilution. The CO₂ molar fractions in standard mixtures prepared by diluting pure CO₂ with air three times deviated by $-0.207 \pm 0.060$ µmol mol⁻¹ on average from the gravimetric

values which were calculated from masses of source materials by evaluating their CO₂ molar fractions based on standard mixtures by diluting the pure CO₂ with the air only once. It indicates that the deviation is larger than a compatibility goal of 0.1 µmol mol⁻¹, which has been recommended by the World Meteorological Organization (WMO). The deviations were consistent with those calculated from the fractionation factors of $0.99968 \pm 0.00010$ and $0.99975 \pm 0.00004$ estimated in mother–daughter transfer

experiment that transfer CO₂/Air mixtures from a cylinder to another evacuated receiving cylinder and by applying the Rayleigh model to the increase in CO₂ molar fractions in source gas as pressure depleted from 11.5 MPa to 1.1 MPa. Both fractionation factors also agree within their uncertainties. Additionally, the mother–daughter transfer experiments showed that the deviation was caused by the fractionation of CO₂ and air in the process of transferring a source gas (a CO₂/Air mixture with a higher CO₂ molar fraction

than that in the prepared gas mixture). The fact that the CO₂ molar fraction weakened significantly as the transfer speed decreased suggested that the main factor of the fractionation could be thermal diffusion. However, experiments exiting a CO₂ in air mixture (CO₂/Air mixture) from a cylinder were





conducted to evaluate the $CO_2$ adsorption to the internal surface of the cylinder. As the cylinder pressure

was reduced from 11.0 to 0.1 MPa, the $CO_2$ molar fractions in the mixture flow leaving from the cylinder

increased the $CO_2$ molar fractions by $0.16 \pm 0.04$ µmol mol$^{-1}$. By applying the Langmuir adsorption-

desorption model to the measured data, the amount of $CO_2$ adsorbed on the internal surfaces of a 10 L

aluminum cylinder when preparing a standard mixture with atmospheric $CO_2$ level was estimated to be

$0.027 \pm 0.004$ µmol mol$^{-1}$ at 11.0 MPa.

**Keywords:** standard mixture, atmospheric $CO_2$, gravimetric method, fractionation

**1 Introduction**

Carbon dioxide ($CO_2$) is an important greenhouse gas that contributes significantly to the radiative forcing

of the atmosphere. Numerous laboratories conduct systematic measurements of atmospheric $CO_2$ to better

understand its sources and sinks. The measurements are typically performed using analyzers calibrated by

working standards traceable to primary standard mixtures determined using manometry and/or gravimetry.

The World Meteorological Organization (WMO) has recommended a compatibility goal of $0.1$ µmol mol$^{-1}$

for $CO_2$ measurements during the Northern Hemisphere (WMO, 2019) to address small and globally

significant gradients over large spatial scales. However, the compatibility goal has not been achieved among

laboratories using their own scales (Tsuboi et al., 2017, Flores et al., 2019), preventing precise evaluation

of sources and sinks of $CO_2$.

Recently, several studies have shown that $CO_2$ adsorbed in the internal surface of a high-pressure cylinder

and desorb from the surface, as the internal pressure decreases (Langenfelds et al., 2005, Leuenberger et

al., 2015, Brewer et al., 2018, Schibig et al., 2018, Hall et al., 2019). These studies also provided a method

to determine the amount of $CO_2$ adsorbed on the internal surface of a cylinder using a "decanting"

experiment to continuously measure the $CO_2$ molar fraction in a $CO_2$ in air mixture ($CO_2$/Air mixture)

exiting a cylinder. For example, Leuenberger et al. (2015) estimated the amount of $CO_2$, expressed as a

fraction of the total gas in a cylinder, to be $0.028$ µmol mol$^{-1}$ at 6 MPa by applying the Langmuir model

(Langmuir, 1918) to the results as 30 L aluminum cylinders were emptied from 6.0 MPa to 0.1 MPa. Schibig



et al (2018) also estimated the amount of $CO_2$ to be $0.0165 \pm 0.0016$ µmol mol$^{-1}$ at 15.0 MPa as 29.5 L aluminum cylinders were emptied from 15.0 MPa to 0.1 MPa. These values cause a small bias in the

gravimetrically assigned $CO_2$ molar fraction in standard mixtures. However, Miller et al. (2015) conducted a series of "mother−daughter" experiments in which they transferred half of a $CO_2$/Air mixture from a "mother" cylinder into an evacuated "daughter" cylinder. They reported that $CO_2$ molar fractions in the mother cylinders were 0.02%−0.03% higher than those in the daughter cylinders. The values were greater than the amounts of adsorbed $CO_2$ estimated by the decanting experiments. According to Hall et al. (2019),

$CO_2$ molar fractions in the mother and daughter cylinders after the mother−daughter experiment were 0.06 µmol mol$^{-1}$ higher and 0.10 µmol mol$^{-1}$−0.13 µmol mol$^{-1}$ lower, respectively, than $CO_2$ molar fractions in the mother cylinders before the transfer. The increasing and decreasing amounts were 5 to 10 times larger than the adsorbed amount estimated from their decanting experiments. They proposed that the detected $CO_2$ change was due to thermal fractionation rather than adsorption of $CO_2$ on the internal surface of a cylinder.

Langenfelds et al. (2005) also assumed diffusive fractionation due to pressure diffusion, thermal diffusion, and effusion were factors that changed $CO_2$ molar fraction observed in $CO_2$/Air mixtures due to gas handling. If the $CO_2$ changes in the transfer are caused by a kinetic process such as the diffusive fractionation, the fractionation factor is considered to be constant regardless of the $CO_2$ molar fraction. In gravimetry, standard mixtures with atmospheric $CO_2$ levels are typically prepared through a multistep

dilution, which involves diluting pure $CO_2$ with air two or three times. Each step of dilution is accomplished by transferring a source gas from a "mother" cylinder into an evacuated "daughter" cylinder and pressurizing it with dilution gas air. The fractionation of $CO_2$ and air (nitrogen, oxygen, argon, and trace impurities other than $CO_2$) is likely to occur in the second and third step dilutions because a $CO_2$/Air mixture with a higher $CO_2$ molar fraction than that in the prepared standard mixture is used as the source gas. The

fractionation process decreases the $CO_2$ molar fraction in the source gas transferred into the daughter cylinder as well as an increase in $CO_2$ molar fraction of the source gas in the mother cylinder because of its consumption. The increase and decrease in $CO_2$ could make the reproducibility of the assigned $CO_2$ molar fractions in the prepared standard mixtures worse. To avoid fractionation in each step dilution, one method

is to gravimetrically prepare standard mixtures by one-step dilution to mix pure $CO_2$ and air directly, as

there is no process to transfer a $CO_2$/Air mixture into another cylinder. Tohjima et al. (2005) developed a

technique to gravimetrically prepare standard mixtures by one-step dilution. However, they did not discuss

fractionation and adsorption that occurs during the multistep dilution process.

To accurately determine the $CO_2$ molar fraction, we must understand the adsorption and fractionation

effects on the preparation process of standard mixtures with atmospheric $CO_2$ levels. Therefore, this study

evaluates the systematic error of $CO_2$'s molar fraction in the standard mixtures prepared by multistep

dilution and the contribution of its factors. $CO_2$ adsorption and fractionation are assumed to depend on the

type and size of a cylinder (Leuenberger, 2015). Although previous studies only evaluated $CO_2$ adsorption

and $CO_2$ and air fractionation in 29.5 L aluminum and 50 L steel cylinders, which were used as working

standards, gravimetric standard mixtures are often prepared in a 10 L aluminum cylinder. Based on

decanting experiments, we evaluate the amount of $CO_2$ adsorbed on the internal surface of a 10 L aluminum

cylinder. The fractionation of $CO_2$ and air in the transfer of $CO_2$/Air mixtures were then evaluated in detail

based on mother–daughter experiments using the cylinders, and the fractionation factor in the transfer of a

source gas was estimated on the basis of the results. Finally, we demonstrated that standard mixtures

gravimetrically prepared by three-step dilutions had a systematic error of $CO_2$ molar fractions by comparing

them with the standard mixtures prepared by one-step dilution.

## 2 Methods

### 2.1 Decanting and mother–daughter experiments

We conducted decanting and mother−daughter experiments to estimate $CO_2$ adsorption in the internal

surface of a cylinder, and the fractionation of $CO_2$ and air during the transfer of a $CO_2$/Air mixture into an

evacuated cylinder.

The decanting experiments were performed using 10 L aluminum cylinders (Luxfer Gas Cylinders, UK)

with a brass diaphragm valve (G-55, Hamai Industries Limited, Japan). The cylinders were evacuated to



~$10^{-4}$ Pa using a turbo molecular pump and pressurized to 11.0 MPa by $CO_2$/Air mixtures with $CO_2$ molar

fractions ranging from 350 µmol $mol^{-1}$ to 450 µmol $mol^{-1}$. The $CO_2$/Air mixtures were decanted using

single stage regulators (Torr 1300, NISSAN TANAKA Co., Japan) attached to the cylinders. The mixture

flow after through the regulator was branched to two ways by T-pieces. The branched flows were controlled

by two mass flow controllers (SEC-Z512MGX 100 SCCM, and 1SLM, Horiba STEC Co., Ltd., Japan);

one controlled flow introduced sample gas into a Picarro G2301 (Picarro, Inc., USA) at a flow rate of 80

ml $min^{-1}$, and the other controlled vent flow at rates of 0 ml $min^{-1}$, 70 ml $min^{-1}$, and 220 ml $min^{-1}$. The

mixtures were emptied from 11.0 MPa to 0.1 MPa at total flow rates of 80 ml $min^{-1}$, 150 ml $min^{-1}$, and 300

ml $min^{-1}$. An absolute pressure gauge of flush Diaphragm type (PPA-33X, KELLER AG, Switzerland)

attached to the regulator was used to measure the pressures in the cylinders. The Picarro G2301 output was

linearly calibrated by a standard mixture containing atmospheric $CO_2$ levels with a standard uncertainty of

less than 0.1 µmol $mol^{-1}$ as the signal was assumed to be zero when the $CO_2$ molar fraction was zero. After

calibrating the Picarro G2301 for 20 min, the process of measuring $CO_2$ in the decanting flow for 100 min

was repeated. The decanting flow was stopped while the Picarro G2301 was calibrated using the standard

mixture.

The mother−daughter experiment was performed using 10 L or 48 L aluminum cylinders (Luxfer Gas

Cylinders, UK) with a brass diaphragm valve. These cylinders were filled with $CO_2$/Air mixtures with $CO_2$

molar fractions ranging from 380 µmol $mol^{-1}$ to 460 µmol $mol^{-1}$ and 3.2 MPa to 13.9 MPa; some of these

mixtures were purchased from a gas supplier (Japan Fine Products, Japan), while others were prepared at

our laboratory. In this experiment, the cylinders containing the mixtures were referred to as the mother

cylinder, while the receiving cylinders into which the mixture was transferred were referred to as the

daughter cylinder. The mixtures were transferred into the evacuated daughter cylinder through a manifold

made of a 1/4-inch o.d. stainless steel line and diaphragm valves (FUDDF-716G, Fujikin Incorporated,

Japan) (Fig. 1a). The manifold was evacuated to ~$10^{-4}$ Pa by a turbo molecular pump after connecting the

mother and daughter cylinder, and then the mixture was expanded from the mother cylinder to the daughter

cylinder by opening the valves of both cylinders. The transfer speed was controlled using the only



diaphragm valve with the daughter cylinders calculated roughly from the transfer time and volume. Both

valves closed immediately after the transfer volume reached the desired level. The transfer volume was

computed using the inner volume and pressure of the daughter cylinder. Molar fractions of $CO_2$ in the

mother cylinders were measured using the Picarro G2301 before starting each experiment, and after each

experiment, those in the mother and daughter cylinders were measured several hours to half a day after the

mixtures were transferred. The Picarro G2301 was calibrated using standard mixtures with atmospheric

$CO_2$ levels before and after each transfer experiment. We also measured $\delta(^{29}N_2/^{28}N_2)$, $\delta(^{34}O_2/^{32}O_2)$,

$\delta(^{32}O_2/^{28}N_2)$, $\delta(^{40}Ar/^{28}N_2)$, and $\delta(^{40}Ar/^{36}Ar)$ in the mother and daughter cylinders using a mass spectrometer

(Delta-V, Thermo Fisher Scientific Inc., USA) to clarify the mechanism(s) of diffusive fractionation during

the mother−daughter experiment based on relationships between the measured elemental and isotopic ratios

(e.g., Langenfelds et al., 2003; Ishidoya et al., 2013). The measurement details of the technique were

provided in Ishidoya and Murayama (2014). The value of $\delta(CO_2/N_2)$ was calculated using the ratio of

$CO_2/N_2$ obtained from Eq. (1) assuming that minor components except $CO_2$ can be ignored ($N_2+O_2,+Ar+$

$CO_2=1$).

$$CO_2/N_2 = \frac{CO_2}{1-CO_2} \times \left( 1 + \frac{O_2}{N_2} + \frac{Ar}{N_2} \right) \qquad\qquad (1)$$

Where $CO_2$ molar fractions measured using Picarro G2301 were used as values of $CO_2$. Atmospheric values

of $N_2$, $O_2$, Ar which are 780894.1 μmol mol$^{-1}$, 209339.1 μmol mol$^{-1}$ and 9334.4 μmol mol$^{-1}$ were used as

values of $N_2$, $O_2$, Ar. The atmospheric values were calculated using the values in previous study (Aoki et

al., 2019)

**2.2 Preparation of standard mixtures**

**2.2.1 Starting Materials for preparation**

Standard mixtures were gravimetrically prepared using the one-step and the three-step dilution in

accordance with ISO 6142-1:2015. Pure $CO_2$ (>99.998 %, Nippon Ekitan Corp., Japan) and G1-grade Air

(Japan Fine Products, Japan) were used as a source gas. The purity of pure $CO_2$ and $N_2$ molar fraction in

the air was determined using a subtraction method in which the sum of molar fractions of impurities was



subtracted from 1 (ISO 19229:2015). Impurities in the source gases were identified and quantified *using*

gas chromatography (GC). A GC with a thermal conductivity detector (TCD) was used to analyze $N_2$, $O_2$,

CH$_4$, and H$_2$ in pure $CO_2$. Ar in the air was analyzed using GC-TCD with an oxygen absorber. A

paramagnetic oxygen analyzer was used to quantify $O_2$ in the Air. A Fourier-transform infrared

spectrometer was used to detect trace amounts of $CO_2$, $CH_4$, and CO in air. A capacitance type moisture

sensor was used to measure $H_2O$ in pure $CO_2$, and a cavity ring-down moisture analyzer was used to

measure $H_2O$ in the Air.

**2.2.2 Balances and weighing sequence**

A 0.8-L aluminum cylinder and a 10-L aluminum cylinder were used for preparing standard mixtures with

atmospheric $CO_2$ levels using a one-step dilution, while a 10-L cylinder was used for preparing a three-step

dilution. The two types of cylinders were weighed using two different balances (mass comparators). One is

AX2005 (Mettler Toledo, Switzerland) used for weighing the 0.8-L cylinder, of which resolution and

maximum load are 0.01 mg and 2 kg, respectively. Another is the XP26003L (Mettler Toledo, Switzerland)

used for weighing the 10-L cylinder, of which the resolution and maximum load are 1 mg and 26 kg

(Matsumoto et al., 2004, Aoki et al., 2019), respectively. The mass measurement of each cylinder, which

was performed in a weighing room controlled at temperature and humidity 26°C ± 0.5°C and 48 % ± 1 %,

respectively, was conducted with respect to a nearly identical reference cylinder to reduce any influence

exerted by zero-point drifts, sensitivity issues associated with the mass comparator, changes in buoyancy

acting on the cylinder, or adsorption effects on the cylinder's surface because of the presence of water vapor

(Alink et al., 2000; Milton et al., 2011). This is performed based on several consecutive weighing operations

in the ABBA order sequence, where "A" and "B" denote the reference and sample, respectively. The

process of loading and unloading the cylinders was automated, and one complete cycle of the ABBA

sequence took five minutes. The mass difference, which was calculated by subtracting the reference

cylinder from the sample cylinder readings, provided the mass reading recorded from the weighing system.

Aoki et al. (2019) reported that the mass reading deviates in relation to temperature differences between





the sample and the surrounding air. In this study, the mass measurement was performed at the sample and

the surrounding areas at the same temperature to reduce the deviation.

**2.2.3 Preparation process by one-step dilution**

Standard mixtures were gravimetrically prepared by mixing pure $CO_2$ and air using stainless steel manifolds

(Fig. 1a, Fig 1b and Fig 1c) in the process shown in Fig. 2a. The pure $CO_2$ cylinder and the 0.8-L aluminum

cylinder were connected at the position of valve 2 (V2) and 6 (V6) to the stainless-steel manifold (Fig. 1b).

The internal surface of stainless-steel manifold were electropolished. The pure $CO_2$ was added to the 0.8-L

aluminum cylinder evacuated to $\sim 5.0 \times 10^{-5}$ Pa via the manifold. Furthermore, we connected the 0.8-L

cylinder and the 10-L cylinder evacuated to $\sim 1.5 \times 10^{-4}$ Pa at the position of the valve 10 (V10) to the

manifold, and then the manifold was evacuated to $\sim 5.0 \times 10^{-5}$ Pa. The valves of the 0.8-L and 10-L

cylinders were opened after V10 was closed, allowing the pure $CO_2$ to expand into the 10-L cylinder. Both

cylinder valves were closed, and then the remaining $CO_2$ in the manifold was moved into the 10-L cylinder

by alternating the pressurization−expansion operation that pressurizes the manifold to ~1.5 MPa with air

and open the valve of the 10-L cylinder. The cylinder was further pressurized to ~10.0 MPa with air using

the manifold shown in Fig. 1c after the $CO_2$ was completely transferred into the cylinder by repeating this

pressurization expansion process 300 times. The $CO_2$ mass filled into the 10-L cylinder was determined by

weighing the 0.8-L cylinder before and after pure $CO_2$ was transferred, whereas the mass of air was

calculated by subtracting the $CO_2$ mass from the difference in the 10-L cylinder mass before and after

transferring pure $CO_2$ and air into the 10-L cylinder.

**2.2.4 Preparation process by three-step dilution**

Fig. 2b shows that the standard mixtures were gravimetrically prepared into the 10-L cylinders by diluting

pure $CO_2$ with air three times in the process. The preparation technique detail was provided in Matsumoto

et al. (2004 and 2008) and Aoki et al. (2019). In the first step dilution, a gas mixture with a $CO_2$ molar

fraction of 65000 µmol mol$^{-1}$, referred to as a 1$^{st}$ gas mixture, was prepared from pure $CO_2$ and air. The

pure $CO_2$ was transferred into the 10-L cylinder evacuated to $1.5 \times 10^{-4}$ Pa, which was then pressurized to 10.0 MPa with air using the manifold shown in Fig. 1c. The masses of pure $CO_2$ and Air were approximately

110 and 1100 g, respectively. In the second step, a gas mixture with a $CO_2$ molar fraction of 5000 μmol $mol^{-1}$, referred to as a 2nd gas mixture, was prepared from the 1st gas mixture and air. The 1st gas mixture was transferred into the 10-L cylinder evacuated to $1.5 \times 10^{-4}$ Pa, which was then pressurized to 10.0 MPa by air. The masses of the 1st gas mixture and air were approximately 100 and 1200 g, respectively. In the third step, a gas mixture with atmospheric $CO_2$ level, referred to as a 3rd gas mixture, was gravimetrically

prepared from the 2nd gas mixture and air. The 2nd gas mixture was transferred into the 10-L cylinder evacuated to $1.5 \times 10^{-4}$ Pa, which was then pressurized to 10.0 MPa with air. The masses of the 2nd gas mixture and air were approximately 100 and 1200 g, respectively. The mass of pure $CO_2$, $CO_2$/Air mixture, and air used as source gases was determined by weighing the cylinder before and after filling each source gas.

**2.2.5 Analysis of standard mixtures**

The gravimetrically prepared standard mixtures (3rd gas mixtures) were measured using the Picarro G2301 equipped with a multiport valve (Valco Instruments Co. Inc., USA) for gas introduction and a mass flow controller (SEC-N112, 100SCCM, Horiba STEC, CO., Ltd, Japan). The output of the Picarro G2301 was calibrated using standard mixtures prepared by the one-step dilution. $CO_2$ molar fractions in the 3rd gas

mixtures were calculated from the calibration line obtained by applying the Deming least-square fit to the measured data.

**3 Result and discussion**

**3.1 Adsorption and fractionation of $CO_2$/Air mixtures**

As described in the introduction, the adsorption of $CO_2$ to a cylinder's internal surface causes a small bias

on the gravimetrically assigned $CO_2$ molar fraction. However, diffusive fractionation in the transfer of source gas is likely to have a larger impact on the $CO_2$ molar fractions than the bias. Therefore, we estimated



the amount of $CO_2$ adsorbed on the internal surface of a 10-L aluminum cylinder, and then fully evaluated the amount of fractionation caused by the transfer of $CO_2$/Air mixtures used as source gases in the evacuated cylinders.

**3.1.1 Amount of $CO_2$ adsorbed on the internal surface of a cylinder**

Previous studies have shown that by applying the Langmuir adsorption-desorption model to the results of decanting experiments, it is possible to determine the amount of $CO_2$ adsorbed on the internal surface of a cylinder. In this method, the amount of $CO_2$ adsorbed on the internal surfaces at the initial pressure of the decanting experiment is expressed as a molar fraction. For example, Schibig et al. (2018) performed a

decanting experiment, emptying 29.5 L aluminum cylinders at a low flow rate of 300 mL min$^{-1}$ and high flow rate of 5 L min$^{-1}$, which is estimated to be $0.0165 \pm 0.0016$ µmol mol$^{-1}$ and $0.043 \pm 0.008$ µmol mol$^{-1}$ at 15.0 MPa, respectively. Leuenberger et al. (2015) also performed the decanting experiment, emptying 30 L aluminum cylinders at a low flow rate of 250 mL min$^{-1}$ and high flow rate of 5 L min$^{-1}$, which is estimated to be 0.028 µmol mol$^{-1}$ at 6.0 MPa and 0.047 µmol mol$^{-1}$ at 9.0 MPa, respectively. The low−flow

decanting experiments indicated that less $CO_2$ was adsorbed on the internal surfaces of cylinders compared to the high-flow decanting experiments. They pointed out that the enrichment of $CO_2$ molar fraction detected in the high flow decanting experiment was related to thermal diffusion and fractionation in the cylinder. Previous studies showed that a low flow decanting experiment is suitable for evaluating the amount of $CO_2$ adsorbed on a cylinder internal surface in the case of 29.5 L and 30 L aluminum cylinders

(Schibig et al., 2018; Leuenberger et al., 2015). It is not known whether this applies to the experiment using 10-L aluminum cylinders. Therefore, we investigated the optimum flow rate to evaluate the adsorbed amount by measuring $CO_2$ molar fraction in a gas mixture exiting in the 10-L cylinder at low flow rates of 80 mL min$^{-}$, 150 mL min$^{-}$, and 300 mL min$^{-1}$ during the decrease in pressure from 11.0 MPa to 0.1 MPa. The deviations in $CO_2$ molar fractions from initial values against relative cylinder pressure ($P/P_0$) at

different flow rates are shown in Fig. 3a. Where $P$ is the actual pressure of the cylinder in MPa and $P_0$ is the initial pressure of the cylinder in MPa before the decanting experiment. The $CO_2$ in the gas mixture



flow increased by $0.16 \pm 0.04$ µmol mol$^{-1}$ as the cylinder pressure decreased from 11.0 MPa to 0.1 MPa.

The standard deviation is indicated by the numbers following the symbol. Unless otherwise noted, the

numbers following the symbol ± represent standard deviation. The increase in $CO_2$ molar fraction is the

same as flow rates of 80 mL min$^{-1}$, 150 mL min$^{-1}$, and 300 mL min$^{-1}$, indicating that the contribution of

thermal fractionation is negligible at a flow rate of 300 mL min$^{-1}$ or less. The amount adsorbed on the

internal surface of the cylinder ($X_{CO_2,ad}$) was calculated using the following equation based on the Langmuir

model as derived by Leuenberger et al. (2015) (Fig. 3b).

$$X_{CO_2,meas} = X_{CO_2,ad} \cdot \left( \frac{K \cdot (P - P_0)}{1 + K \cdot P} + (1 + K \cdot P_0) \cdot \ln\left( \frac{P_0 \cdot (1 + K \cdot P)}{P \cdot (1 + K \cdot P_0)} \right) \right) + X_{CO_2,initial}$$

(2)

Where $X_{CO_2,ad}$ is expressed as the $CO_2$ molar fraction multiplied by the occupied adsorption sites at

pressure $P_0$. $X_{CO_2,meas}$, corresponding to the measured molar fraction. $X_{CO_2,initial}$ is the $CO_2$ molar fraction

measured in the cylinder at a pressure $P_0$. $K$ is the ratio of the adsorption and desorption rate constants, and

its unit is MPa$^{-1}$. $X_{CO_2,ad}$ and $K$ was obtained from the least square fit to the results. These experiments

were performed seven times, and the average of $X_{CO_2,ad}$ was $0.027 \pm 0.004$ µmol mol$^{-1}$, corresponding to

0.030 mL standard temperature and pressure (STP) at 11.0 MPa or 1.2 micromoles or $7.3 \times 10^{17}$ molecules.

There was no difference in the values of $X_{CO_2,ad}$ in range of $CO_2$ from 350 to 450 µmol mol$^{-1}$. The ratio of

the adsorption of $CO_2$ to $CO_2$ in the cylinder is $0.008 \% \pm 0.001 \%$ at a unit of mole. The inner diameter of

0.16 m, length of 0.56 m, and the internal surface area are roughly calculated to be 0.32 m$^2$ because our

cylinders have an outer diameter of 0.18 m. The occupied area of $CO_2$ adsorbed on the internal surface was

estimated to be 0.06 m$^2$, assuming a molecule diameter of 3.4 Å, corresponding to approximately 20 % of

the inner area by a monolayer of adsorbed $CO_2$ molecules. The adsorbed amount by the third step dilution

was considered when $CO_2$ molar fractions in 3$^{rd}$ gas mixture were gravimetrically determined in the

following section because the adsorption of $CO_2$ causes a small bias of $CO_2$ molar fraction in a cylinder.

However, the amount was neglected in the case of the 1$^{st}$ and 2$^{nd}$ gas mixtures. This is because the $CO_2$

molar fraction is significantly higher than the atmospheric $CO_2$ level by 10 and 100 times or more. In the Langmuir model, the increase rate of the amount adsorbed on the internal surface with increasing partial pressure of $CO_2$ becomes small as the molar fraction of $CO_2$ increases. The adsorbed amount is assumed to

be lower than the adsorption ratio of 0.008 % ± 0.001 % in the case of the 1st and 2nd gas mixtures with a high molar fraction of $CO_2$.

### 3.1.2 Mother–daughter experiment

The fractionation of $CO_2$ and air in the transfer of a gas mixture with atmospheric $CO_2$ level has can be caused by not only the diffusive process but also the adsorption process. The fractionation of the adsorption

process is assumed to be caused by the increase in the amount of $CO_2$ adsorbed on the internal surface according to the cooling of the mother cylinder in the transfer of the gas mixture. The temperature decreases in the mother cylinders observed in our mother–daughter experiments was 2−8 K. Leuenberger et al. (2015) identified the temperature dependence in the amount of $CO_2$ adsorbed on the internal surface of an aluminum cylinder to be in a range from −0.0002 μmol mol$^{-1}$ K$^{-1}$ to −0.0003 μmol mol$^{-1}$ K$^{-1}$. This

corresponded to the decrease in 0.0004 μmol mol$^{-1}$−0.0024 μmol mol$^{-1}$ for $CO_2$ molar fractions in the mixtures transferred from the mother cylinder, which is significantly lower than the decrease in $CO_2$ molar fraction detected in the daughter cylinders. Therefore, the fractionation of $CO_2$ and air is predicted to occur by the diffusive fractionation process based on the three types of diffusion, i.e., pressure diffusion, thermal diffusion, and effusion as described by Langenfelds et al. (2005) and Moore et al. (1962). The pressure

diffusion is driven by a pressure gradient. The diffusion caused heavier molecules to be preferentially accumulated in the region of higher pressure. The thermal diffusion is driven under a temperature gradient. Heavier molecules are preferentially accumulated in the colder region. The effusion is known as Knudsen diffusion. Gas molecules escaping from a pressurized vessel through a tiny orifice are subject to molecule effusion. This Knudsen diffusion occurs when the size of the orifice is small compared to the mean free

path among molecular collisions, indicating that most collisions are with the walls of the orifice. Neither



lighter nor heavier molecules preferentially escape from the orifice because the rate of effusion is inversely proportional to the square root of the mass.

Mother–daughter experiments of gas mixtures with atmospheric $CO_2$ levels were performed in twelve sets using 48-L and 10-L aluminum cylinders as mother cylinders and 10-L aluminum cylinders as daughter

cylinders. The experiments were conducted at different mother cylinder's pressure, transfer gas amount, and transfer gas speed, to understand the contributions of pressure diffusion, thermal diffusion, and effusion in the transfer of the gas mixture. The mother cylinder's pressure determines the pressure gradient between mother and daughter cylinders. The pressure gradient also changes according to the transferred gas amount. The transfer gas speed determines the thermal gradient. These gradients drive pressure and thermal

diffusion. Additionally, the molecular mass of $CO_2$ and air contributes to pressure diffusion, thermal diffusion, and effusion.

The mother–daughter experimental results are summarized in Table 1. Here, $CO_2$ molar fractions in the daughter cylinders were corrected by the amount of $CO_2$ absorbed on the internal surface based on the value of $0.027 \pm 0.004$ µmol mol$^{-1}$ determined by the decanting experiment. The dependence of $CO_2$ molar

fractions in the daughter cylinders relative to transfer volume, cylinder pressure, and transfer speed is shown in Fig. 4. The closed circles in Fig. 4 represent values in the transfer speed of more than 19 L min$^{-1}$, whereas open triangles represent values in the transfer speed of less than 3 L min$^{-1}$. All $CO_2$ molar fractions in the mixtures transferred into the daughter cylinders decreased from the $CO_2$ molar fraction before the transfer of the mixtures are shown in Fig. 4. The decrease in $CO_2$ molar fractions mixtures for the daughter cylinders

was $0.122 \pm 0.040$ µmol mol$^{-1}$ on average at a transfer speed of more than 19 L min$^{-1}$, whereas the decrease in $CO_2$ molar fractions for daughter cylinders from initial values became significantly small as $0.036 \pm 0.027$ µmol mol$^{-1}$ ($0.008 \% \pm 0.006 \%$) on average when the mixtures were transferred at an extremely slow transfer speed of less than 3 L min$^{-1}$. The decreased values at the transfer speed of more than 19 L min$^{-1}$ agreed with previous values of 0.10 and 0.13 µmol mol$^{-1}$ reported by Hall et al. (2019), who reported

that the decrease could be related to thermal diffusion. However, the mixtures for all mother cylinders provided higher $CO_2$ molar fractions than before the transfer of the mixture, contrary to the daughter

cylinders. The amount of substance ($n$) for increased and decreased $CO_2$ in the mother and daughter

cylinders was computed from the change amount of $CO_2$ molar fraction ($c_{CO_2}$) to evaluate the mass balance

of $CO_2$ corresponding to increase and decrease in $CO_2$ molar fractions, which is related to the initial value

before the transfer of the mixture, and the cylinder volume ($V$) and pressure ($p$) in the daughter cylinder

using the ideal gas low; $n = c_{CO_2} \times p \times V/(R \times T)$. Where R and T express gas constant (0.082057 L

atm $K^{-1}$ $mol^{-1}$) and gas temperature (298 K), respectively. The mass balance between the increase and

decrease was consistent within uncertainties in each experiment (Table 1), indicating that the changes of

$CO_2$ were caused by the diffusive fractionation rather than $CO_2$ adsorption.

As shown in Fig. 4, the $CO_2$ decrease does not depend on the transfer volume and initial pressure of the

mother cylinder, but it does become significantly weaker as the transfer speed decreases. The fact that the

amount of $CO_2$ molar fractions decreased was constant regardless of the transfer volume, indicates that the

fractionation factor did not change at the beginning and end of the transfer. These results also indicate that

the fractionation is likely to be caused by thermal diffusion rather than pressure diffusion and effusion

because the transfer speed determines the thermal gradient. The influence of thermal diffusion on the molar

fraction of $CO_2$ was constant in the case of the transfer speed of more than 19 L $min^{-1}$, but it was

significantly suppressed by the transfer of the mixture at a lower transfer speed. Therefore, controlling the

transfer speed of a source gas may allow for the preparation of standard mixtures with accurately and

precisely atmospheric $CO_2$ levels even when $CO_2$ standard mixtures are prepared by multistep dilutions.

However, it is difficult to transfer source gases at the transfer speed presented in this experiment because

the speed is much lower than the normal transfer speed in the preparation process of the standard mixtures.

We must acquire a technique to control the transfer speed of source gas.

A fractionation factor ($\alpha$) in the transfer of a source gas was estimated from the results for the transfer speed

of more than 19 L $min^{-1}$ because the transfer speed of a source gas in the actual preparation of standard

mixtures is significantly larger than 19 L $min^{-1}$. $CO_2$ molar fraction in the gas mixture in the cylinder ($X_{out}$)

is modified by the fractionation factor to the ratio in the cylinder as the following equation.



$$X_{out} = \alpha X_0. \tag{3}$$

Where $X_0$ is the initial $CO_2$ molar fractions. The fractionation factor ($\alpha$) was estimated to be $X_{out}/X_0 =$ 0.99968 ± 0.00010 using the values in Table 1. If a standard mixture with a $CO_2$ molar fraction of 400 μmol $mol^{-1}$ is prepared by a three-step dilution, the $CO_2$ molar fraction in the standard mixture is predicted to decrease 0.252 ± 0.082 μmol $mol^{-1}$ by the fractionation effect in the second and third step dilutions. Additionally, the $CO_2$ molar fraction in a source gas ($X$) can be expressed using pressure ($P$) and initial

pressure ($P_0$) of the source gas by the Rayleigh fractionation model.

$$\frac{X}{X_0} = \left(\frac{P}{P_0}\right)^{\alpha-1} \tag{4}$$

According to equation (4), the $CO_2$ molar fraction in the source gas is estimated to be 1.00076 ± 0.00024

against an initial value with a decrease in pressure from 11.0 to 1.0 MPa. This value corresponds to the increase in 0.30 ± 0.09 μmol $mol^{-1}$ from the initial value in a standard mixture with a $CO_2$ molar fraction of 400 μmol $mol^{-1}$ prepared from the source gas.

We also measured different molecular pairs, $^{32}O_2/^{28}N_2$, $^{40}Ar/^{28}N_2$, and $CO_2/N_2$, and the same molecular pairs, $^{29}N_2/^{28}N_2$, $^{34}O_2/^{32}O_2$, and $^{40}Ar/^{36}Ar$ to understand the diffusive effects on the fractionating process. The

relationship of the deviations of $\delta(^{32}O_2/^{28}N_2)$, $\delta(^{40}Ar/^{28}N_2)$, $\delta(CO_2/N_2)$, $\delta(^{34}O_2/^{32}O_2)$, and $\delta(^{40}Ar/^{36}Ar)$ with deviations of $\delta(^{29}N_2/^{28}N_2)$ in the daughter cylinders relative to their mother cylinders are shown in Fig. 5. The black line represents the values obtained from the mother–daughter experiment using 10 and 48 L cylinders. The red dotted line, blue dotted line, and black dotted line represent the theoretical values of pressure diffusion, thermal diffusion, and effusion, respectively, which were calculated using the equations

provided by Langenfelds et al. (2005). Red solid lines represent the deviations due to thermal diffusion experimentally estimated by Ishidoya et al. (2013, 2014). Here, note that the deviation of the experimental thermal diffusion for $\delta(CO_2/N_2)$ has large uncertainty and requires further experiments. The deviations of


the experimental thermal diffusion for $\delta(CO_2/N_2)$, $\delta(^{32}O_2/^{28}N_2)$, and $\delta(^{40}Ar/^{28}N_2)$ were larger than their theoretical deviations, whereas the deviations of experimental thermal diffusion for $\delta(^{34}O_2/^{32}O_2)$ and

$\delta(^{40}Ar/^{36}Ar)$, which are not shown in Fig. 5, were consistent with the theoretically calculated values. The deviations of $\delta(CO_2/N_2)$ in the daughter cylinders relative to their mother cylinders were close to the experimental deviation although they were significantly larger than the theoretical deviations. The fact that the deviations of $\delta(CO_2/N_2)$ are close to the experimental thermal diffusion, indicating that the fractionations occurred by thermal diffusion. The indication was also supported by the results, which show

that the deviations of different pairs, $\delta(^{32}O_2/^{28}N_2)$ and $\delta(^{40}Ar/^{28}N_2)$ were consistent with those of the experimental thermal diffusion rather than the theoretical deviations although there was significant variability. Additionally, the deviations of the same molecular pairs, $\delta(^{34}O_2/^{32}O_2)$ and $\delta(^{40}Ar/^{36}Ar)$ were also theoretically and experimentally used for thermal diffusion although they cannot be discriminated from pressure diffusion and effusion because of the variability of deviations. These results show that thermal

diffusion can be the main factor in the fractionation of $CO_2$ and air. Unfortunately, our measurement values were scattered to evaluate the factors causing fractionation in detail. Therefore, the deviations of $\delta(CO_2/N_2)$ can also be caused by other factors except for thermal diffusion. For example, there may be unknown fractionation mechanism(s), depending on the molecular size like the close off fractionation assumed to occur in rock-in-zone of firn (e.g., Severinghaus and Battle, 2006). Additionally, the adsorption theory may

have not been sufficiently understood. Additional studies must clarify the mechanisms of the fractionation of $CO_2$ and air in more detail.

**3.2 Comparation between one-step dilution and three-step dilutions**

In the previous section, we determined the fractionation factor in the transfer of a source gas to be 0.99968 ± 0.00010. This indicates that the $CO_2$ molar fraction in gravimetrically prepared standard mixture with

atmospheric $CO_2$ level has a systematic error by the fractionation in the dilution process by the transfer of $CO_2$/Air mixture used as a source gas to an evacuated daughter cylinder in the second and third step dilution. Two types of experiments were conducted to confirm the systematic error. One evaluated the fractionation





in the second and third step dilutions based on the increase in $CO_2$ molar fractions in 1st and 2nd gas mixtures due to the fractionation with their consumption. Another demonstrated that $CO_2$ molar fractions in 3rd gas

mixtures deviate from their gravimetric values by measuring 3rd gas mixtures based on standard mixtures prepared by one-step dilution, which can avoid the fractionation.

Two series of standard mixtures were prepared by one-step dilution to determine $CO_2$ molar fractions in the 3rd gas mixtures used in the two experiments. The $CO_2$ molar fractions were corrected on the basis of the adsorption of $CO_2$ to the internal surface using the $X_{CO_2,ad}$ of $0.027 \pm 0.004$ µmol mol$^{-1}$. Four standard mixtures were prepared as the first series to evaluate the fractionation in the second and third steps, and the $CO_2$ molar fractions were $390.687 \pm 0.077$ µmol mol$^{-1}$, $402.253 \pm 0.078$ µmol mol$^{-1}$, $415.452 \pm 0.080$ µmol mol$^{-1}$, and $426.602 \pm 0.082$ µmol mol$^{-1}$. Five standard mixtures were prepared as the second series to demonstrate the deviations of $CO_2$ molar fractions in the 3rd gas mixtures in which the $CO_2$ molar fractions were $390.599 \pm 0.078$ µmol mol$^{-1}$, $399.807 \pm 0.094$ µmol mol$^{-1}$, $402.724 \pm 0.094$ µmol mol$^{-1}$, $406.021 \pm 0.094$ µmol mol$^{-1}$, and $419.618 \pm 0.098$ µmol mol$^{-1}$. The numbers following the symbol ± denote expanded uncertainty, which was mainly associated with the mass of $CO_2$ and air. The molar mass of air also contributes to the uncertainty of the $CO_2$ molar fraction because the composition of the air is different among individual cylinders of the same gas manufacturer. For example, $O_2$ molar fractions in the air, which our laboratory uses ranges from 208000 µmol mol$^{-1}$ to 209600 µmol mol$^{-1}$. This difference causes the $CO_2$ molar fraction to deviate by 0.09 µmol mol$^{-1}$. Therefore, the molar fractions of $N_2$, $O_2$, and Ar in the air used in this experiment were determined based on standard mixtures composed of $N_2$, $O_2$, Ar, and $CO_2$. Ar molar fractions were determined to range from 9300 µmol mol$^{-1}$ to 9360 µmol mol$^{-1}$ using GC-TCD, and their largest standard uncertainty was 6 µmol mol$^{-1}$, whereas $O_2$ molar fractions were determined to range from 208804 µmol mol$^{-1}$ to 209276 µmol mol$^{-1}$ using the paramagnetic $O_2$ analyzer and their largest standard uncertainty was 6 µmol mol$^{-1}$. $N_2$ molar fractions in the air were calculated by subtracting the Ar and $O_2$ molar fractions from 1. The results of the first and second series measured using the Picarro G2301 are shown in Fig. 6a. The line represents the Deming least-square fit to the data. The residuals from the line are shown in Fig. 6b. The error bar is expressed as the expanded uncertainty of gravimetric values. The



residual ranges from −0.014 µmol mol$^{-1}$ to 0.008 µmol mol$^{-1}$ for the first series and from −0.057 µmol

mol$^{-1}$ to 0.054 µmol mol$^{-1}$ for the second series. The measured molar fractions were consistent with the

line within the expanded uncertainties.

To evaluate the increase in $CO_2$ molar fraction in 2$^{nd}$ gas mixture as the source gas, six reference mixtures

(3$^{rd}$ gas mixtures) with approximately 400 µmol mol$^{-1}$ were prepared from a common 2$^{nd}$ gas mixture,

which had a gravimetric value of 5022.46 ± 0.18 µmol mol$^{-1}$ for $CO_2$ in the process shown in Fig. 7a. The

number following the symbol ± denotes the expanded uncertainty. The pressure of the 2$^{nd}$ gas mixture used

for the preparation of the 3$^{rd}$ gas mixtures was 11.5 MPa, 9.7 MPa, 8.05 MPa, 4.2 MPa, 2.75 MPa, and 1.1

MPa. The increase in $CO_2$ molar fractions in the 2$^{nd}$ gas mixture was evaluated by measuring the 3$^{rd}$ gas

mixtures using the Picarro G2301 based on the first series because the increase in the 2$^{nd}$ gas mixture

directly reflects them in the 3$^{rd}$ gas mixtures prepared from the 2$^{nd}$ gas mixture. These contributions are

negligible to the increase because all cylinders act similarly, although the fractionation in the transfer of the

2$^{nd}$ gas mixture into the daughter cylinder and adsorption of $CO_2$ to the internal cylinder surface also affect

$CO_2$ molar fraction in the 3$^{rd}$ gas mixture. The relationship of the deviations from the gravimetric values in

the 3$^{rd}$ gas mixtures and the pressure of the 2$^{nd}$ gas mixture is shown in Fig. 8a. The vertical axis is expressed

as the deviation values to subtract the measured values from the gravimetric values for the 3$^{rd}$ standard

mixtures. The error bars represent the expanded uncertainties calculated based on combining the standard

uncertainty of the measurement with that of the gravimetric values for the standard mixtures prepared by a

three-step dilution. The deviations increased by 0.25 ± 0.10 µmol mol$^{-1}$ as the pressure decreases from 11.5

to 1.1 MPa, and it agrees with the increased value of 0.30 ± 0.10 µmol mol$^{-1}$ predicted from Eq. (4) using

the fractionation factor of 0.99968 ± 0.00010 determined in section 3.1. However, we estimated the

fractionation factor in the third step dilution by applying the Rayleigh fractionation model [the Eq. (4)] to

increase the decrease in inner pressure, as shown in the solid line in Fig. 8a. The estimated fractionation

factor was 0.99975 ± 0.00004, which was consistent with the fractionation factor of 0.99968 ± 0.00010

estimated in section 3.1. This consistency indicates that the fractionation detected in the mother–daughter

experiment also occurs in the transfer of a source gas in the preparation process of the 3$^{rd}$ gas mixtures.



The fractionation of $CO_2$ and air is also assumed to occur in the second step dilution in which the 1st gas

mixture composed of $CO_2$ and air was transferred to the evacuated cylinder. We evaluated the fractionation

based on the change in the deviations from the gravimetric values in 3rd gas mixtures prepared using the

process shown in Fig. 7b. Two 3rd gas mixtures with a $CO_2$ molar fraction of approximately 400 μmol mol$^{-1}$

were prepared from two 2nd gas mixtures, which were prepared using a common 1st gas mixture having a

$CO_2$ molar fraction of 65164.9 ± 1.9 μmol mol$^{-1}$. The 2nd gas mixtures had $CO_2$ molar fractions of 5022.46

± 0.18 μmol mol$^{-1}$ and 4824.67 ± 0.35 μmol mol$^{-1}$, which were prepared from the 1st gas mixture at a

pressure of 7.8 and 0.8 MPa. The 2nd gas mixtures were used only for the preparation of the 3rd gas mixtures.

The number following the symbol ± denotes the expanded uncertainty. The $CO_2$ molar fractions in the 3rd

gas mixtures were determined using the Picarro G2301, which is based on the first series. The contributions

of the fractionation of $CO_2$ in the daughter cylinder and adsorption of $CO_2$ to increase the depletion in inner

pressure were canceled because of the reasons described in the previous paragraph. The relationship of the

deviations in the measured values from the corresponding gravimetric values and pressure of the 1st gas

mixture is shown in Fig. 8b. The solid and dotted lines in Fig. 8b represent the Rayleigh model line, which

was calculated based on the fractionation factor of 0.99975 ± 0.00004 and 0.99968 ± 0.00010. The error

bars represent the expanded uncertainties calculated based on the combination of standard uncertainty of

the measurement with that of the gravimetric values for the 3rd gas mixtures. The deviations increased by

0.16 ± 0.10 μmol mol$^{-1}$ as the pressure decreased from 7.8 MPa to 0.8 MPa. Both lines agree with the

deviations within the uncertainties. The results mean that the fractionation factor in the second step dilution

is equivalent to the fractionation factor in the third step dilution. This means that fractionation occurs

regardless of the $CO_2$ molar fraction of a source gas.

Finally, we demonstrated that the $CO_2$ molar fraction in the 3rd gas mixture deviated from its gravimetric

value according to the fractionation factors described above. In this demonstration, four 3rd gas mixtures

for atmospheric $CO_2$ levels were newly prepared by three-step dilutions. The increase in $CO_2$ molar

fractions in the 1st and 2nd gas mixtures with their consumption were corrected on the basis of the decrease

in their pressures from the initial values. The decreases in $CO_2$ molar fractions by the adsorption of $CO_2$ to





the internal surface for 3rd gas mixtures were corrected based on the $X_{CO_2,ad}$ of $0.027 \pm 0.004$ µmol mol⁻¹. These corrections allow for extracting only the deviations from gravimetric values caused by fractionation in the transfer of 1st and 2nd gas mixtures. The $CO_2$ molar fractions in the 3rd gas mixtures were measured using the Picarro G2301 based on the second series. The measured values of $CO_2$ molar fractions were

calculated based on the calibration line obtained by applying the Deming least-square fit to the measured values. The deviations were calculated by subtracting the gravimetric values from the measured values in the 3rd gas mixtures. The error bars represent the expanded uncertainties of the gravimetric values. The deviations were $-0.207 \pm 0.060$ µmol mol⁻¹ on average. The deviation was dropped between $-0.252 \pm 0.082$ µmol mol⁻¹ and $-0.200 \pm 0.032$ µmol mol⁻¹ calculated using the fractionation factor of $0.99968 \pm$

$0.00010$ and $0.99975 \pm 0.00004$, and it was consistent with both values within their uncertainty. This indicates that the fractionation of $CO_2$ and air occurs according to our estimated fractionation factor in each dilution process. $CO_2$ molar fractions standard mixtures prepared by multistep dilutions were identified as systematic error according to the fractionation of $CO_2$ and air. Therefore, we must consider the fractionation when determining $CO_2$ molar fraction in standard mixtures gravimetrically prepared by multistep dilutions.

**4 Conclusion**

Adsorption and fractionation $CO_2$ and air were used to evaluate systematic deviations during the preparation of a standard mixture with atmospheric $CO_2$ levels. Decanting experiments were performed to evaluate the amount of $CO_2$ adsorbed on the internal surface of a 10-L aluminum cylinder during the preparation of $CO_2$/Air mixtures at the atmospheric level. The amount of adsorbed $CO_2$ was determined to be $0.027 \pm$

$0.004$ µmol mol⁻¹ at 11.0 MPa, resulting in a small bias in the gravimetric value. The mother–daughter experiments were performed to understand the fractionation of $CO_2$ and air when a $CO_2$/Air mixture used was transferred into an evacuated cylinder as a source gas. $CO_2$ molar fractions in the mother and daughter cylinders increased and decreased, respectively, indicating that fractionation causes not only a decrease in $CO_2$ molar fraction in the prepared standard mixture but also an increase in $CO_2$ molar fraction in a source

gas. The decrease of $CO_2$ mole fractions in the daughter cylinders does not depend on the transfer volume

nor on the initial pressure of the mother cylinder, while it clearly weakens with decreasing transfer speeds, since thermal diffusion is the main factor of fractionation. The fractionation factor in the transfer of the $CO_2$/Air mixture was $0.99968 \pm 0.00010$, indicating that the $CO_2$ molar fraction decreased by $0.032\ \% \pm 0.010\ \%$ by transfer of a source gas and the $CO_2$ molar fraction in a source gas increases by $0.30 \pm 0.10$

$\mu$mol mol$^{-1}$ as the inner pressure decreased from 11.5 MPa to 1.1 MPa. We found that the fractionation factor was $0.99975 \pm 0.00004$ by analyzing the increase in $CO_2$ molar fraction in a source gas during the actual preparation of standard mixtures. We demonstrated that $CO_2$ molar fractions in standard mixtures by three-step dilutions decreased by $-0.207 \pm 0.060$ $\mu$mol mol$^{-1}$, which is greater than the compatibility goal of 0.1 $\mu$mol mol$^{-1}$, from gravimetric values based on source gas fractionation. The decrease was between

the values calculated using the fractionation factors of $0.99976 \pm 0.00004$ and $0.99968 \pm 0.00010$. The fractionation caused the $CO_2$ molar fraction to increase and decrease. The reproducibility of $CO_2$ molar fractions in gravimetric standard mixtures will suffer as a result. We must consider the change in $CO_2$ molar fraction caused by fractionation when gravimetrically preparing standard mixtures in multistep dilutions.

**Code availability**

**Data availability.** The data presented in this article are available upon request to Nobuyuki Aoki (aoki-nobu@aist.go.jp).

**Author contribution.** NA designed the study. NA performed the experiment and drafted the paper. SI carried out measurement by a mass spectrometry. NM helped with the preparation of standard mixtures.



SM helped with determination of $CO_2$ molar fraction. All were actively involved with the final version of the paper.

**Competing interests.** The authors declare that they have no conflict of interest.

**Disclaimer**

**Acknowledgments**

This study was partly supported by the Global Environment Research Account for National Institutes of the Ministry of the Environment, Japan (grant nos. METI1454 and METI1953) and the JSPS KAKENHI Grant Number 19K05554.

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

Table 1. Results of the mother–daughter experiment on 10-L and 48-L aluminum cylinders. $CO_2$/air mixtures at atmospheric level were transferred from 10-L or 48-L aluminum cylinders (mother) to 10-L aluminum cylinders 610     (daughter) at different mother cylinder's pressure, transfer volume, and transfer speed.

| Cylinder | | Size (L) | Pressure [a] | | molar fraction [b] | | Drift [c] | | Transfer [d] |
| | number | | Before (MPa) | After (MPa) | Before (µmol/mol) | After (µmol/mol) | Amount (µmol) | Molar fraction (µmol/mol) | Speed (L/min) |
|---|---|---|---|---|---|---|---|---|---|
| Mother | CPC00878 | 10 | 9.8 | 4.4 | 379.138 | 379.322 | $3.15 \pm 0.73$ | 0.18 | 62 |
| Daughter | CPC00875 | 10 | | 4.5 | | 379.035 | $-1.80 \pm 0.74$ | $-0.10$ | |
| Mother | CPD00092 | 10 | 10.5 | 4.8 | 458.611 | 458.715 | $1.96 \pm 0.79$ | 0.10 | 211 |
| Daughter | CPD00093 | 10 | | 4.4 | | 458.488 | $-2.11 \pm 0.73$ | $-0.12$ | |
| Mother | CPD00076 | 10 | 4.1 | 2.0 | 378.103 | 378.243 | $1.09 \pm 0.33$ | 0.14 | 27 |
| Daughter | CPB28688 | 10 | | 2.0 | | 377.982 | $-0.94 \pm 0.33$ | $-0.12$ | |


| | | | | | | | | | |
|---|---|---|---|---|---|---|---|---|---|
| Mother | CPD00069 | 10 | 13.5 | 8.0 | 377.523 | 377.602 | 2.46 ± 1.32 | 0.08 | 216 |
| Daughter | CPD00072 | 10 | | 4.5 | | 377.334 | −3.31 ± 0.74 | −0.19 | |
| Mother | CPD00070 | 10 | 13.2 | 7.8 | 377.936 | 378.026 | 2.73 ± 1.29 | 0.09 | 24 |
| Daughter | CPD00074 | 10 | | 5.1 | | 377.751 | −3.66 ± 0.84 | −0.19 | |
| Mother | CPB16349 | 10 | 8.8 | 7.0 | 419.319 | 419.350 | 0.84 ± 1.16 | 0.03 | 54 |
| Daughter | CPC00484 | 10 | | 1.7 | | 419.135 | −1.21 ± 0.28 | −0.18 | |
| Mother | CPD00069 | 10 | 6.6 | 5.6 | 377.602 | 377.635 | 0.72 ± 0.93 | 0.03 | 19 |
| Daughter | CPD00072 | 10 | | 0.8 | | 377.463 | −0.43 ± 0.13 | −0.14 | |
| Mother | CQB15834 | 48 | 14.5 | 8.6 | 376.876 | 376.950 | 12.49 ± 7.10 | 0.07 | 167.7 |
| Daughter | CPD00072 | 10 | | 8.1 | | 376.781 | −2.96 ± 1.33 | −0.09 | 55.2 |
| | CPD00074 | 10 | | 8.0 | | 376.792 | −2.60 ± 1.31 −8.44 ± 2.33 | −0.08 | 54.5 |
| | CPD00073 | 10 | | 8.5 | | 376.788 | −2.88 ± 1.40 | −0.09 | 57.9 |
| Mother | CQB15808 | 48 | 13.9 | 8.5 | 377.200 | 377.255 | 9.18 ± 5.01 | 0.05 | 291.6 |
| Daughter | CPD00070 | 10 | | 8.3 | | 377.129 | −2.29 ± 1.37 | −0.07 | 99.6 |
| | CPD00069 | 10 | | 7.8 | | 377.095 | −3.20 ± 1.29 −8.69 ± 2.32 | −0.11 | 93.6 |
| | CPD00076 | 10 | | 8.2 | | 377.100 | −3.20 ± 1.36 | −0.10 | 98.4 |
| Mother | CPB31362 | 10 | 4.13 | 3.3 | 441.693 | 441.722 | 0.37 ± 0.54 | 0.03 | 2.8 |
| Daughter | CPB16311 | 10 | | 0.86 | | 441.641 | −0.17 ± 0.14 | −0.05 | |
| Mother | CPB31362 | 10 | 3.2 | 1.6 | 406.184 | 406.223 | 0.24 ± 0.26 | 0.04 | 1.1 |





| Daughter | CPB16311 | 10 | | 1.5 | | 406.180 | $-0.02 \pm 0.25$ | $-0.004$ | |
|---|---|---|---|---|---|---|---|---|---|
| Mother | CPB28912 | 10 | 8.5 | 4.5 | 419.853 | 419.908 | $0.95 \pm 0.74$ | 0.05 | 2.2 |
| Daughter | CPB16463 | 10 | | 4.0 | | 419.801 | $-0.80 \pm 0.66$ | $-0.05$ | |

[a] Pressures were measured using the pressure gauge attached the regulator.

[b] $CO_2$ molar fractions in mother and daughter cylinders were measured after several hours to half of a day of transferring the mixtures. These values have a measurement uncertainty of 0.030 μmol/mol.

[c] The change in the amount of substance ($n$) for $CO_2$ were computed from the change in the amount of $CO_2$ molar fraction ($c_{CO_2}$), the cylinder volume (V) and pressure (p) in the daughter cylinder using the ideal gas low; $n = c_{CO_2} \times p \times V/(R \times T)$. Numbers following the symbol $\pm$ denote the standard uncertainties calculated based on the measurement uncertainty.

[d] Transfer speeds were roughly computed by dividing transfer volume by transfer time.



(a)

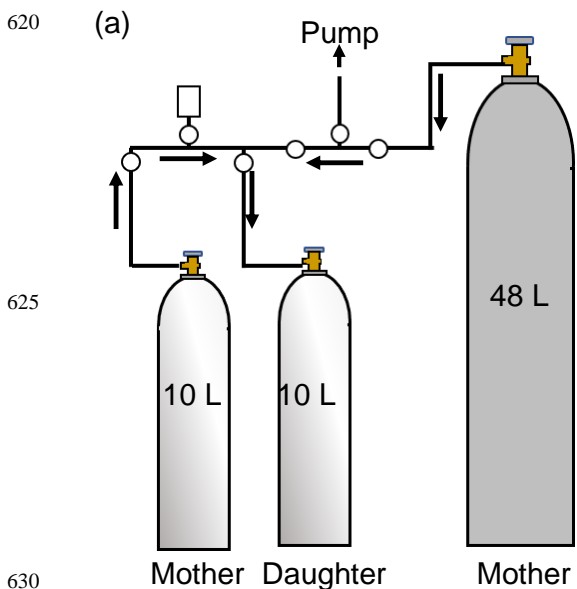

b)

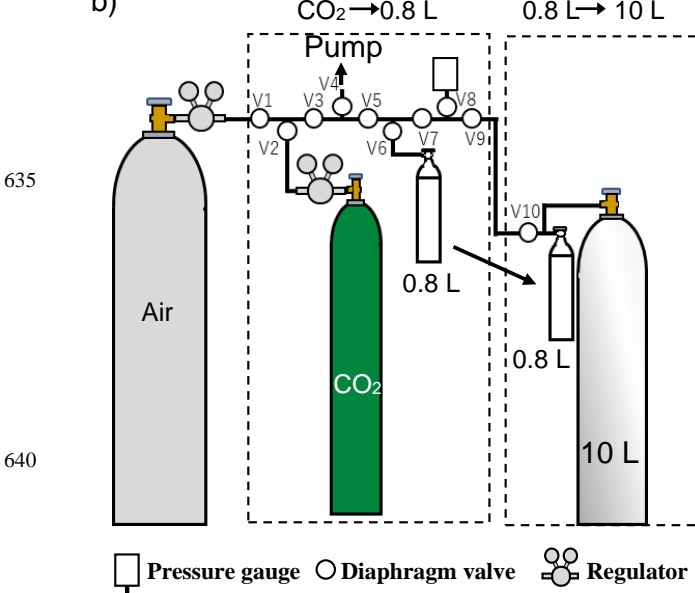





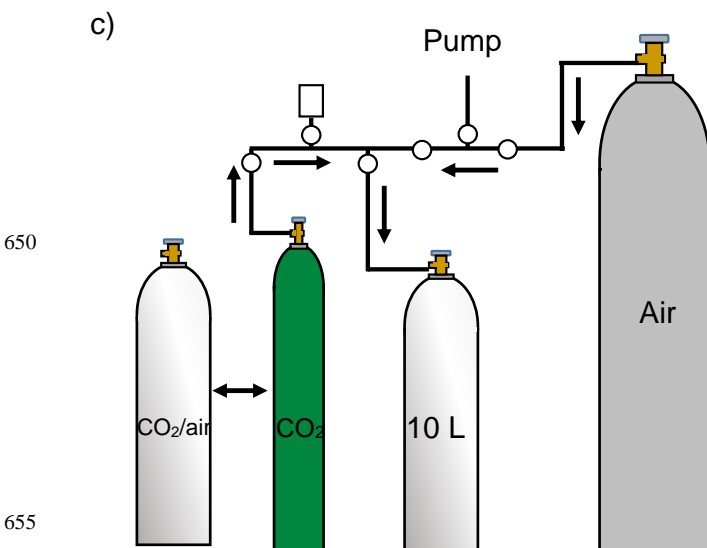

650

655

Figure 1 (a) Schematic of the manifold used to transfer a $CO_2$/air mixture from a mother cylinder to a daughter cylinder in a mother–daughter experiment, (b) the manifold used to transfer pure $CO_2$ to a 0.8-L aluminum cylinder and from a 0.8-L aluminum cylinder to a 10-L aluminum cylinder for preparing a standard mixture via one-step dilution and (c) 660    the manifold used to transfer source gas (pure CO2 or a CO2/air mixture) and dilution gas (air).





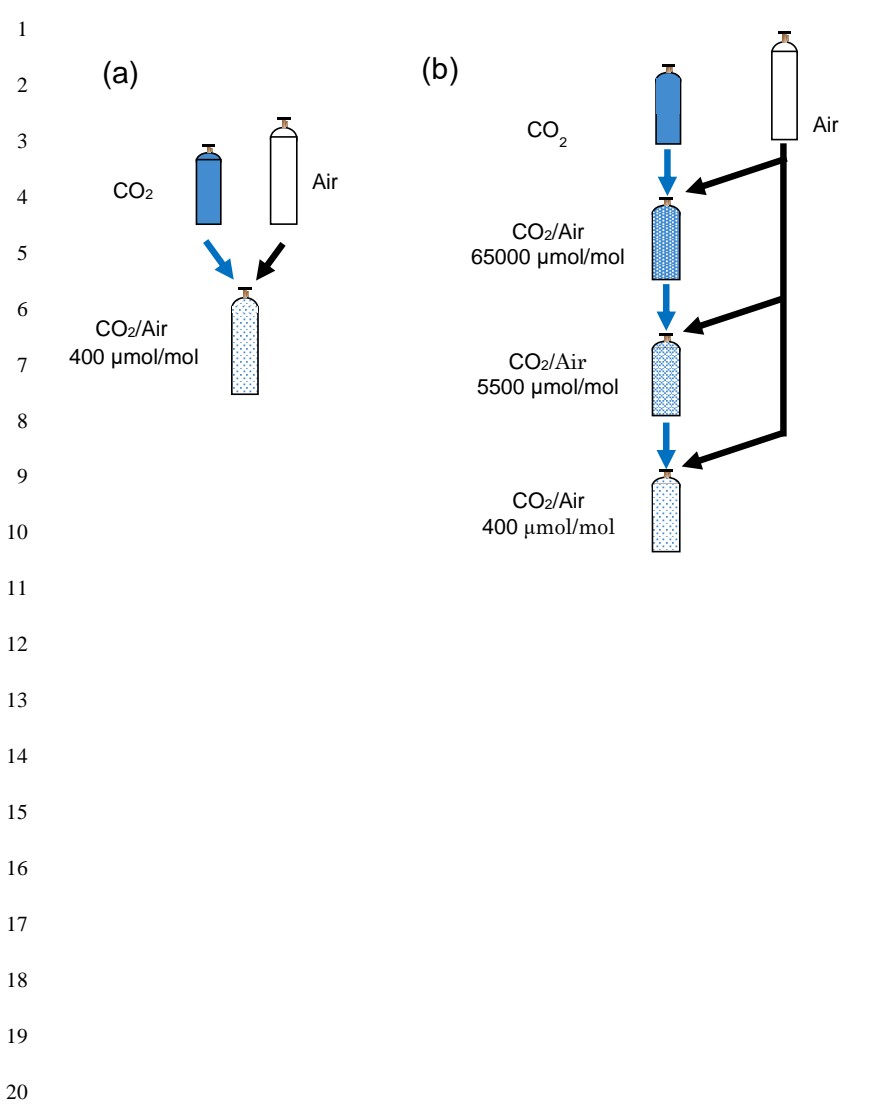

Figure 2 (a) Preparation process of standard mixtures under atmospheric $CO_2$ level via one-step dilution. (b) Preparation process of 3[rd] gas mixtures under atmospheric $CO_2$ level via three-step dilutions.



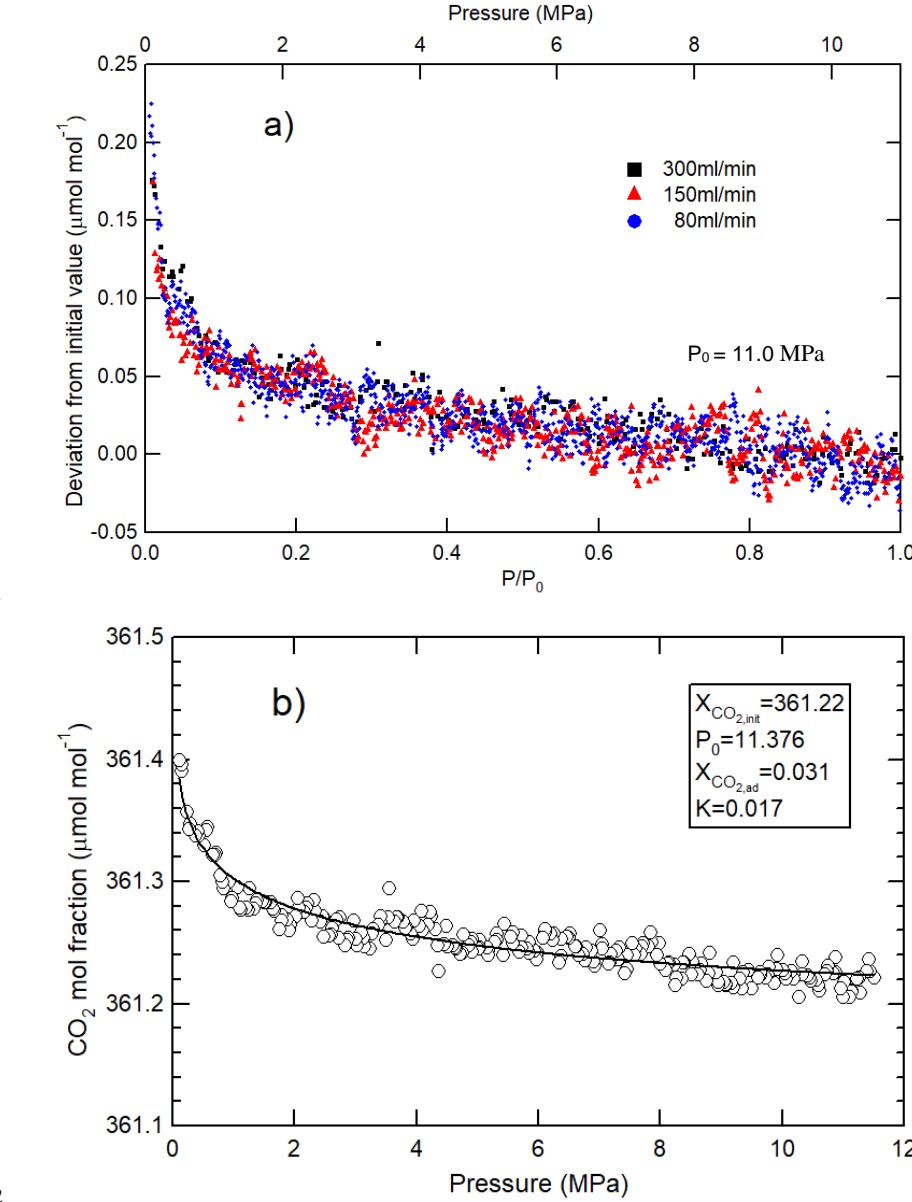

Figure 3 (a) Change of the $CO_2$ molar fractions from initial value in $CO_2$/air mixtures under atmospheric $CO_2$ level

against relative pressure as the cylinder was emptied at the flow rates of 80 mL min$^{-1}$, 150 mL min$^{-1}$, and 300 mL

min$^{-1}$ from 11.0 MPa to 0.1 MPa. (b) Typical results obtained by applying the Langmuir model to the change of $CO_2$

molar fractions from initial value in $CO_2$/air mixture as the cylinder was emptied from 11.0 MPa to 0.1 MPa.



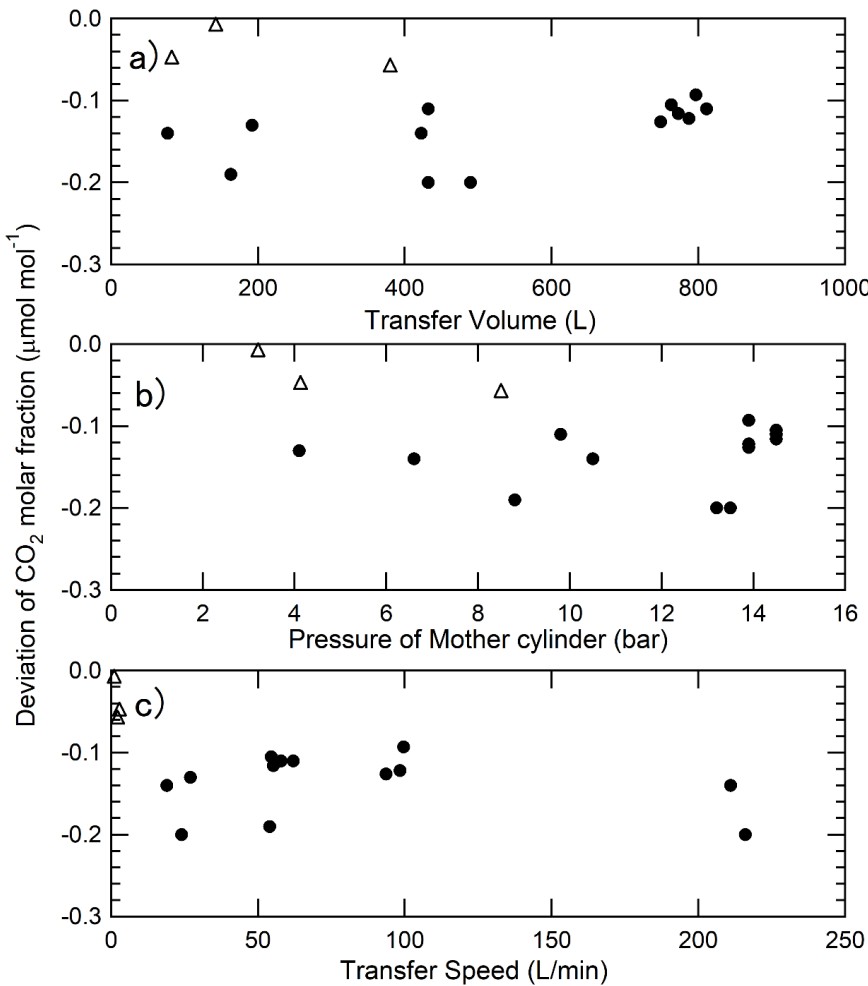

Figure 4 Deviations of $CO_2$ molar fractions in daughter cylinders from initial values against the mother cylinder's

pressure, and transfer volume and transfer speed when the $CO_2$/air mixtures under atmospheric level were transferred

from the mother cylinder to the daughter cylinder at different mother cylinder's pressures, transfer gas amounts, and

transfer gas speeds. The filled circles represent the results measured at a transfer speed of more than 19 L min[-1],

while the open triangles represent the results measured at a transfer speed of less than 3 L min[-1]

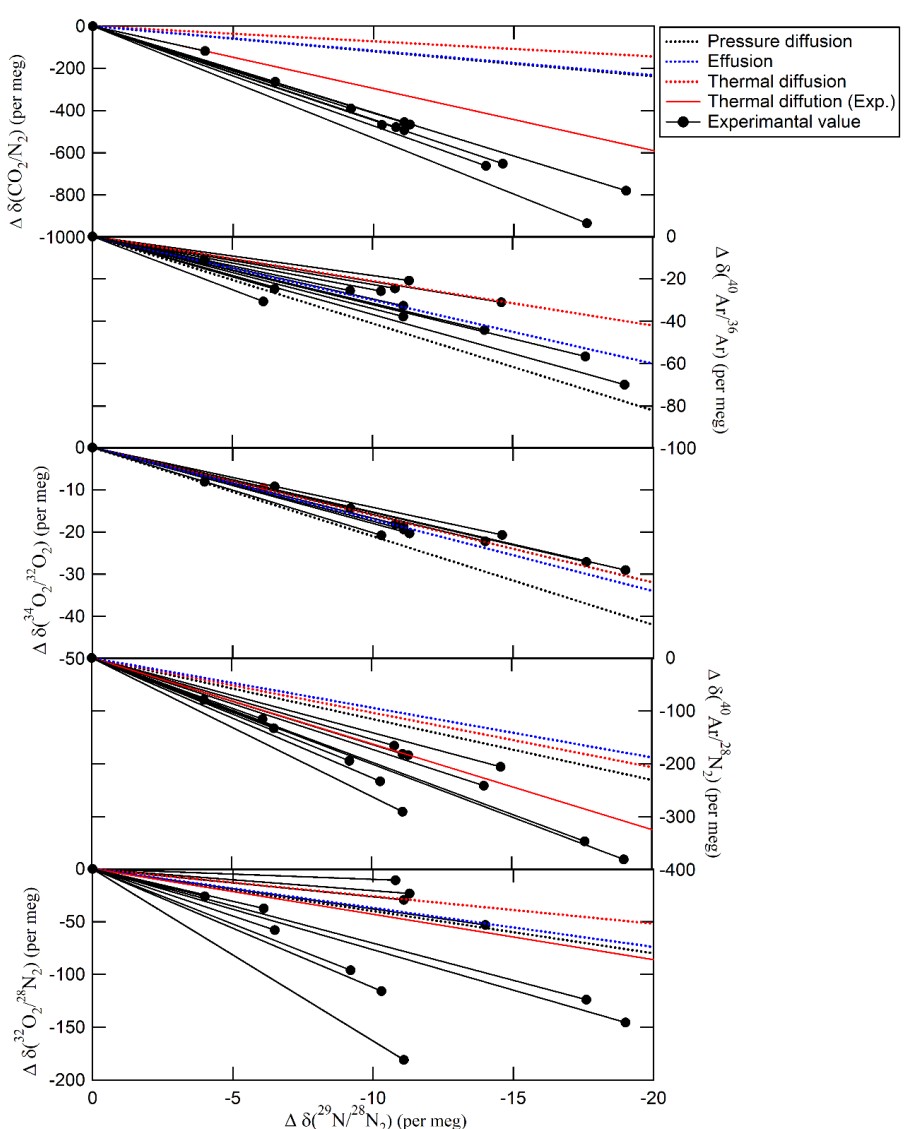

Figure 5 Relationship between the deviations of $\delta(^{44}CO_2/^{28}N_2)$, $\delta(^{40}Ar/^{36}Ar)$, $\delta(^{34}O_2/^{32}O_2)$, $\delta(^{40}Ar/^{28}N_2)$, and

$\delta(^{32}O_2/^{28}N_2)$ with the deviations of $\delta(^{29}N_2/^{28}N_2)$ in the daughter cylinders relative to their mother cylinders after the

$CO_2$/air mixtures under atmospheric level were transferred from the mother cylinder to the daughter cylinder. The red,

blue ,and black dotted lines represent the theoretical values of pressure diffusion, thermal diffusion, and effusion,

respectively, (Langenfelds et al. 2005). The red solid lines represent the deviations due to thermal diffusion,

experimentally estimated by Ishidoya et al. (2013, 2014).





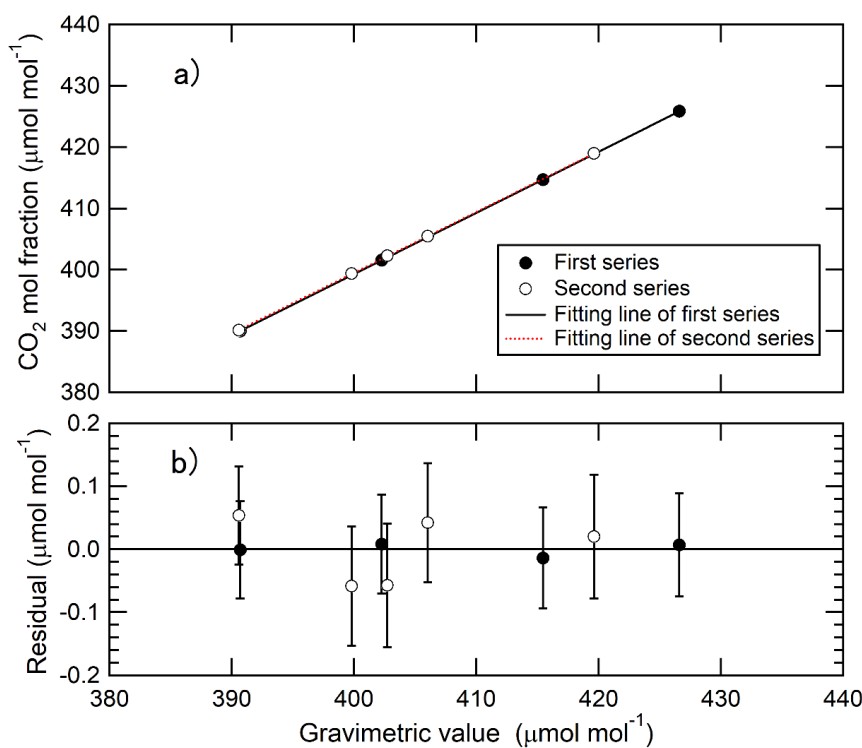

3    Figure 6 (a) Relationships between the measured $CO_2$ molar fractions and the gravimetric values for two series of

4    standard mixtures prepared via one-step dilution. (b) Residuals from the Deming least-square fit shown in (a).



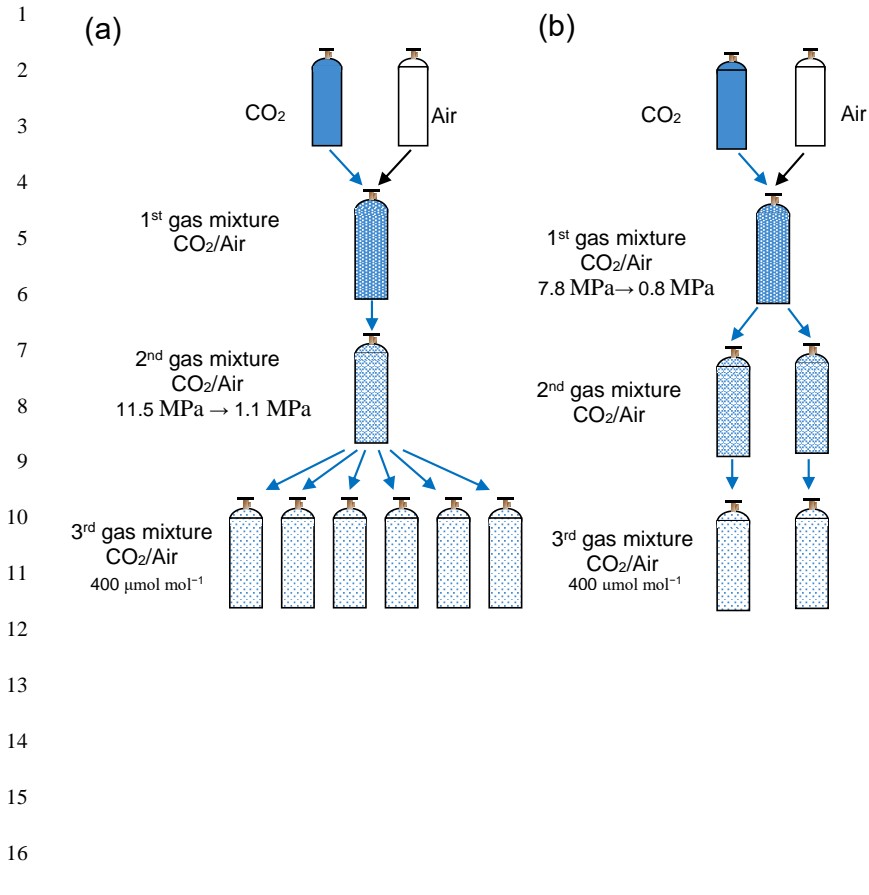

Figure 7 (a) Preparation process of the 3rd gas mixtures under atmospheric $CO_2$ level via three-step dilutions to evaluate the fractionation in the third step and (b) second step dilutions.



3 Figure 8 (a) Deviations of the measured $CO_2$ molar fractions from the gravimetric values against the pressure of the $2^{nd}$

4 gas mixture. $CO_2$ molar fractions determined on the basis of the standard mixtures prepared via one-step dilution. The

5 solid line represents the Rayleigh model fit for the plots. (b) Deviations of the measured $CO_2$ molar fractions from the

6 gravimetric values against the pressure of the $1^{st}$ gas mixture. The $CO_2$ molar fractions determined on the basis of the

7 standard mixtures prepared via one-step dilution. The solid and dotted lines represent the Rayleigh model fit based on

8 the fractionation factor of $0.99975 \pm 0.00004$ and $0.99968 \pm 0.00010$.





1    Re

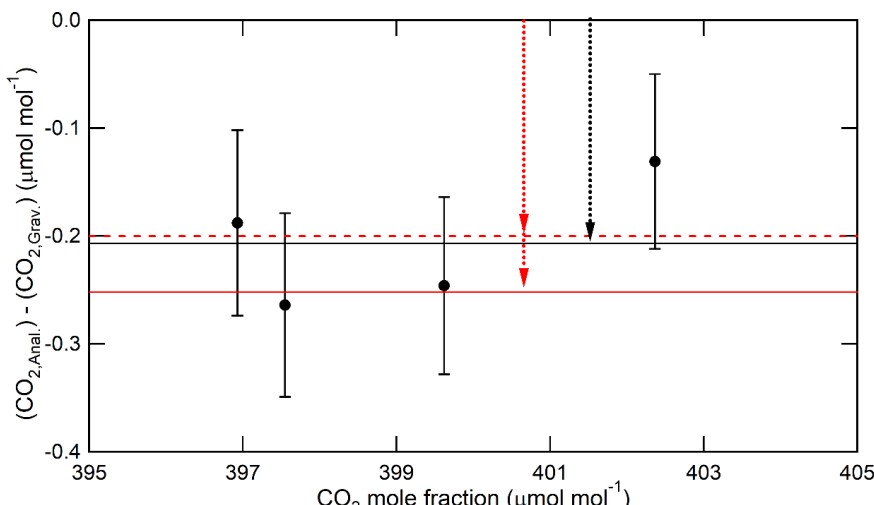

3    Figure 9 Deviations of the measured values from the gravimetric values of $CO_2$ molar fractions in the standard mixtures

4    (3rd gas mixtures) prepared via three-step dilutions. The measured values were calculated from the calibration line

5    obtained by applying the Deming least square fit to the measured data. The black line represents the average value of

6    the deviations. The red solid and dotted lines represent the values calculated using fractionation factors of 0.99968 ±

7    0.00010 and 0.99975 ± 0.00004, respectively. The red and black arrows represent the deviation of $CO_2$ molar fraction

8    in the 3rd gas mixtures according to the fractionation of $CO_2$ and air.

