# Peer review of "Influence of CO2 adsorption on cylinders and fractionation of CO2 and air during the preparation of a standard mixture"

_Atmospheric Measurement Techniques, 2022_

## Referee Comment (RC2)

Review of "Influence of adsorption of CO2 on cylinder and fractionation of CO2 and air during preparation of a standard mixture" by N. Aoki et al, AMTD.

**General:**

The manuscript presents new updated  $CO_2$  concentration (and auxiliary) measurements regarding fractionations associated with adsorption/desorption on metal surfaces as well as fractionations during decanting experiments (mother-daughter, dilution experiments). These fractionations, in particular the latter, are relevant for assigning concentration and isotope values as best as possible as it significantly changes the concentration values. Aoki et al., did a thorough experimental study combined with explanations using models that has been used earlier on. Their results clearly show that care must be taken during dilution and decanting (mother-daughter) experiments.

This manuscript deserves publication in AMT after the manuscript has been checked for English language shortcomings as mainly addressed by Reviewer 1 and additionally outlined below.

**Minor points:**

- Title: consider changing to: Influence of CO2 adsorption on cylinders and the fractionation of CO2 and air during the production of a standard mixture
- Abstract: We conducted a study to fully understand carbon dioxide (CO2) adsorption I do not know whether you should write it in such an absolute manner, consider skipping fully or exchange it with better.
- Abstract: The CO2 molar fractions in standard mixtures prepared by diluting pure CO2 with air three times deviated by -0.207 ± 0.060 µmol mol-1 on average from the gravimetric values which were calculated from masses of source materials by evaluating their CO2 molar fractions based on standard mixtures by diluting the pure CO2 with the air only once. This sentence is difficult to understand, consider splitting it up.
- Abstract: rewrite: When the cylinder pressure was reduced from 11.0 to 0.1 MPa, the CO2 mole fractions in the mixture stream exiting the cylinder increased by  $0.16 \pm 0.04$  µmol mol-1.
- Intro: However, the compatibility goal has not been achieved among laboratories using their scales (Tsuboi et al., 2017, Flores et al., 2019), preventing precise evaluation of sources and sinks of CO2.

Here, I agree with Reviewer 1. Your conclusion is indeed misleading as the accuracy within the WMO GAW network does not play role as all the values needs to be reported on the same scale. The accuracy of the scale itself is of second-order. Your cited references document differences among different scales in use. Please reformulate this part.

- Line 105-106: The mixture flow after through the regulator was branched to two ways by Tpieces. T, rephrase to "After flowing through the regulator, the mixture flow was branched in two ways by T-pieces.
- Line 141: ....(N2+O2,+Ar+CO2=1). Use ....(N2+O2+Ar+CO2=1)

- Line 507-510:The fractionation factor in the transfer of the CO2/Air mixture was  $0.99968 \pm 0.00010$ , indicating that the CO2 molar fraction decreased by  $0.032 \% \pm 0.010 \%$  by transfer of a source gas and the CO2 molar fraction in a source gas increases by  $0.30 \pm 0.10 \mu$ mol mol-1 as the inner pressure decreased from 11.5 MPa to 1.1 MPa. rephrase this sentence.
- Fig. 4: refer to subgraphs a), b) and c) in the Figure legend

---

## Author Response (AR1)

Response to reviewer 1

**General:**

The manuscript presents new updated CO2 concentration (and auxiliary) measurements regarding fractionations associated with adsorption/desorption on metal surfaces as well as fractionations during decanting experiments (mother-daughter, dilution experiments). These fractionations, in particular the latter, are relevant for assigning concentration and isotope values as best as possible as it significantly changes the concentration values. Aoki et al., did a thorough experimental study combined with explanations using models that has been used earlier on. Their results clearly show that care must be taken during dilution and decanting (mother-daughter) experiments.

This manuscript deserves publication in AMT after the manuscript has been checked for English language shortcomings as mainly addressed by Reviewer 1 and additionally outlined below.

Response: We appreciate the reviewer's thoughtful review and constructive comments. All the comments have been addressed in the revised manuscript, and the responses to each comment are given below.

**Minor points:**

Title: consider changing to: Influence of CO2 adsorption on cylinders and the fractionation of CO2 and air during the production of a standard mixture

**Response:** we revised title according to your comment.

Abstract: We conducted a study to fully understand carbon dioxide (CO2) adsorption
I do not know whether you should write it in such an absolute manner, consider skipping fully or exchange it with better.

**Response:** we revised the sentence to "We conducted a study to evaluate carbon dioxide ($CO_2$) adsorption". (P1, L10)

Abstract: The CO2 molar fractions in standard mixtures prepared by diluting pure CO2 with air three times deviated by $-0.207 \pm 0.060$ $\mu$ mol mol$-1$ on average from the gravimetric values which were calculated from masses of source materials by evaluating their CO2 molar fractions based on standard mixtures by diluting the
pure CO2 with the air only once.
This sentence is difficult to understand, consider splitting it up.

**Response:** We revised the sentence to "It became clear that the $CO_2$ molar fractions in

standard mixtures prepared by diluting pure $CO_2$ with air three times deviated by $-0.207 \pm 0.060$ $\mu$ mol mol$^{-1}$ on average from the gravimetric values." (P1, L12-14)

Abstract: rewrite: When the cylinder pressure was reduced from 11.0 to 0.1 MPa, the CO2 mole fractions in the mixture stream exiting the cylinder increased by $0.16 \pm 0.04$ $\mu$ mol mol-1.
**Response:** We revised the sentence according to your comment. (P1, L26 - P2, L28)

Intro: However, the compatibility goal has not been achieved among laboratories using their scales (Tsuboi et al., 2017, Flores et al., 2019), preventing precise evaluation of sources and sinks of CO2.
Here, I agree with Reviewer 1. Your conclusion is indeed misleading as the accuracy within the WMO GAW network does not play role as all the values needs to be reported on the same scale. The accuracy of the scale itself is of secondorder. Your cited references document differences among different scales in use.
Please reformulate this part.
**Response:** we reformulated this part according to your comment. (P2. L39-43)

Line 105-106: The mixture flow after through the regulator was branched to two ways by Tpieces. T, rephrase to "After flowing through the regulator, the mixture flow was branched in two ways by T-pieces.
**Response:** We revised the sentence according to your comment. (P55, L107-108)

Line 141: ⋯.(N2+O2,+Ar+CO2=1). Use ⋯.(N2+O2+Ar+CO2=1)
**Response:** We revised the sentence according to your comment. (P6, L151)

Line 507-510:The fractionation factor in the transfer of the CO2/Air mixture was $0.99968 \pm 0.00010$, indicating that the CO2 molar fraction decreased by $0.032 \% \pm 0.010 \%$ by transfer of a source gas and the CO2 molar fraction in a source gas increases by $0.30 \pm 0.10$ $\mu$ mol mol$-1$ as the inner pressure decreased from 11.5 MPa to 1.1 MPa.
rephrase this sentence.
**Response:** We reconstructed the sentences. (P21, L509-515)

Fig. 4: refer to subgraphs a), b) and c) in the Figure legend
**Response:** we revised Fig.4 according to your comment.

Response to reviewer 2

RC1: 'Comment on amt-2022-41', Anonymous Referee #2, 11 May 2022 reply
The manuscript describes experimental work to quantify effects that affect the mole fraction of $CO_2$ in standard air mixtures prepared by gravimetric methods. This certainly is a highly relevant topic within the scope of AMT. The topic has been addressed already by other papers that are adequately referenced. The experimental work has been performed thoroughly, the results of many individual experiments are very consistent and such add valuable quantitative data on the adsorptive loss of $CO_2$ on high pressure cylinder surfaces and fractionation effects when transfering gas mixtures from one cylinder to another for the purpose of preparing diluted samples. In previous studies of the investigation of such adsorptive effects results it was not always unambiguous if those results might also have been impacted by thermal fractionation. The very good agreement of several decanting experiments performed in this manuscript at different low flow rates yielding similar adsorbed quantities of $CO_2$ at cylinder walls as others have also have published provides further confirmation of the quantitative relevance of this effect.

The results of the second series of eperiments on fractionation during gas mixture transfer from one cylinder to an evacuated second cylinder (mother-daughter and dilution experiments) show clearly that the $CO_2$ mole fraction is altered by the process. It is shown that this effect is quantitatively very reproduceable with the experimental setup used for these experiments. This setup normally involves high flow rates for the gas transfer that are probably the cause for the disturbance. Three mother-daughter experiments have been conducted at lower transfer speeds resulting in a lower fractionation. However, this mode of operation has not been further employed at the panel utilized apparently is not made for such low flows. At this point the description of the experimental setup has not been sufficiently detailed to understand this. Isotope ratio mass spectrometry has been used to study the relationships in fractionation of various isotope or molecule pairs (e.g. $29N_2/28N_2$; $34O_2/32O_2$; $Ar/N_2$; $O_2/N_2$;..) and compare these measured relationships with literature results as an indicator for the kind of diffusive process causing the fractionation. These measurements apparently are not trivial and neither could a convincing conclusion be drawn from these data nor appears the discussion of these results very substantial.

There is a problem with the language being not sufficiently clear in several sentences throughout the manuscript that requires a thorough revision. For some key sentences I have provided suggestions below but this is certainly not exhaustive.

Concrete points that I feel the authors need to further elaborate or correct before final publication are listed here below.

**Response:** We appreciate the reviewer's thoughtful review and constructive comments. All the comments have been addressed in the revised manuscript, and the responses to each comment are given below.

lines 40-44: The conclusion drawn here is misleading and therefore should not remain. The quoted WMO compatibility goal is part of WMO recommendations that request to refer to the WMO $CO_2$ mole fraction scale. By using this approach the absolute accuracy of the scale is of less importance and not a major contribution to the limit of the WMO GAW network compatibility. However, the two comparison studies that are refered to (Tsuboi et al, 2017 and Flores et al 2019) both comprise participating laboratories that mostly report on different scales. In the Japanese study it appears that for each lab the consistency of offsets of three comparison samples relative to NIES is very good for all the labs and the agreement between AIST and TU (both reporting on the TU2010 scale) is within the WMO compatibility goal. The comparison results of the CCQM Key Comparison (Flores et al 2019) provides the result of measurements performed at the BIPM of standard gases provided by the participating NMIs and the WMO Central Calibration Laboratory. Both references therefore do not provide any information on the compatibility of measurements by different laboratories relating to the WMO scale that might limit an evaluation of the carbon cycle.

**Response:** we reformulated this part according to your comment (P2, L39-43).

l.81: please insert reference: ".. into another cylinder (Hall et al. 2019)"

**Response:** We inserted the reference according to your comment(P4, L80-81).

p.5 l. 120: " some of these mixtures were purchased from a gas supplier (Japan Fine Products, Japan), while others were prepared at our laboratory.". This does not state if the gas mixtures are synthetic (which becomes clear later in the manuscript) or based on real ambient air. This should be stated explicitly here. What is confusing is the statement made at a later point (p. 6 l. 143) that "Atmospheric values of $N_2$, $O_2$, Ar ... were used as values of $N_2$, $O_2$, Ar.". This does not appear valid for synthetic mixtures.

**Response:** We stated information of the gas mixtures (P5, L123-124).

p.5/6 l. 127f: I guess that I do not understand the following sentence and it is not in clear agreement with Fig 1a: "The transfer speed was controlled using the only diaphragm valve

with the daughter cylinders calculated roughly from the transfer time and volume."

It appears that the full pressure of the mother cylinder is applied to the system with the evacuated daughter cylinder at the other end. This should cause a dramatically large flow entailing big temperature changes in the Mother and Daughter cylinders. There are several valves depicted in the figure that the gas is passing through. I do not know the type of diaphragm valve so what restriction it does provide. The one pressure gauge in Fig. 1a is downstream of diaphragm valves from the Mother cylinders and upstream of the diaphragm valve of the Daughter cylinders so it is not clear to me how it is checked when the transfer volume has been reached. How is the transfer time then measured? How long is the transfer typically taking? In Fig. 1a there are two cylinders named Mother and one Daughter. Please revise to make it clear.

**Response:** These parts had not been clear. We explained the transfer method of the mixtures in detail by revision of sentences and Figure 1a(P5, L131 - P6, L136).

p 6. l 141: It would be helpful if the equation 1 could be explicitly derived.

**Response:** We revised the equation 1 and the sentence about the equation 1(P6, L152).

p 6. l. 149: Could you please specify what is G1-grade air?

**Response:** We revised the sentence to "purified Air (G1-grade ,Japan Fine Products, Japan)" (P7, L158-159).

p.7 l. 158: Was there any significant humidity detectable in the air?

**Response:** $H_2O$ concentration in the air was under 0.1 ppm.

p.8 l. 184ff: It is not explained why the 0.8L cylinder is moved from V6 to V10. Is it in order to avoid any $CO_2$ getting lost in the the valves V6-V9? If so, how has been assured that no additional $CO_2$ that might have adsorbed in valves V1 - V7 is later flushed into the 10 L cylinder during the 300 pressure expansion cycles with the dilution air?

**Response:** We revised Figure1b and sentences (P8, L192-197). We had confirmed that additional $CO_2$ that have adsorbed in valves was not detected by measuring Air flushed in the manifold after pure $CO_2$ was transferred although we did not describe it in the paper.

p.8 l. 197: Was this preparation done using the set-up displayed in Fig. 1a?

**Response:** It was done using the set-up displayed in Fig. 1a. We added "by using the manifold shown in Fig 1a." at the end of the sentence (P9, L209).

p.9 l. 215f: "The output of the Picarro G2301 was calibrated using standard mixtures prepared by the one-step dilution." In section 2.2.3 it is not specified how many one-step dilution standards are prepared in parallel. This information is provided at a later stage but number of calibration points and the CO2 mole fraction range covered should also be mentioned here.
**Response:** We added the information after "The output of the Picarro G2301 was calibrated using standard mixtures prepared by the one-step dilution." (P10, L230-232).

p.9 l. 223: "However, diffusive fractionation in the transfer of the source gas is likely to have a larger impact on the CO2 molar fractions than the bias by the sorption processes." This appears like an anticipation of the results the study is concluding with. It also relates to the large flow rates employed in this experimental set-up.
**Response:** we revised "Furthermore, transfer of CO2/air mixture changes CO2 molar fractions by about 0.10 $\mu$mol mol$-1$. This allow transfer of source gases to have a larger impact on the CO2 molar fractions than the bias on adsorption process. " (P10, L236-238)

p.11 l. 250: Omit the first sentence ("The standard deviation is..")
**Response:** We omitted the sentence (P11, L264).

p.12 l. 297-299: This sentence needs to be corrected as follows: " Lighter molecules preferentially escape from the orifice relative to heavier ones because the rate of effusion is inversely proportional to the square root of the mass".
**Response:** We removed the sentence because sentences about effusion were reformulated (P11, L292-300).

p.13 l. 306: Is the diffusive process quick enough to cause a fractionation because of the pressure gradient between Mother and Daughter cylinder entail very high bulk flow rates?
**Response:** We think that the fractionation occurs in the mother cylinders. Mother-daughter experiments from horizontal mother cylinders into the daughter cylinders suggested that thermal diffusion was main factor of the fractionation. Therefore, we added the experiment results in Fig.4 and revised the text (P13, L310-P14, L327).

p.13 l. 307f: I would think this sentence: "Additionally,.." should be omitted, as all these diffusive processes are based on molecular mass differences.
**Response:** We omitted the sentence according to your comment.

p.13 l. 313f / Fig. 4 / Fig. 1b: The description of the panel used for the mother-daughter experiments depicted in Fig. 1b does not make clear how the transfer speed is controlled but Fig. 4 clearly indicates that in the experimental series transfer speed and transfer volume were independently controlled. Therefore, please do clarify in Fig. 1a or section 2.1 by what means (which type of valves) the flow can be regulated.

**Response:** We revised Fig. 1a and explained how the transfer speed is controlled in the section 2.1 (P6, L131-L136).

p.14 l. 333: The statement "..,but it becomes significantly weaker as the transfer speed decreases" suggests a continuous relationship which the data do not show. Rather change to "..but it becomes significantly smaller below flow rates of 19 L/min"

**Response:** We revised the sentence according to your comment (P15, L353).

p.14 l. 336: Why would you consider Knudsen diffusion to play a role? The orifice of the valves certainly will exceed the free path length of molecular collisions. Omit "effusion".

**Response:** As you describe, the orifice will exceed the free path length of molecular collisions. We omitted "effusion".

p.14 l. 344: This sentence poses again the question: how has the flow been regulated to perform the experiments resulting in the transfer speeds < 3L/min (last three data points of Table 1 and open triangles in Fig. 4) ?

**Response:** We explained how the transfer speed is controlled in the section 2.1.

p.15 l. 333: Does the statement "..using the values in Table 1." include alle the values (that is what is written) or only values with transfer speeds ≥ 19 L/min (that would make more sense to me, as apparently the fractionation factor is not constant below some threshold between 3 and 19 L/min)?

**Response:** We thought that the statement "···using the values in Table 1" include only values with transfer speeds 19 L/min. We revised the sentence to "···using only values with transfer speeds 19 L/min in Table 1" (P16, L372)

p.15 l. 372: As stated above, Knudsen diffusion ("effusion") will not be of relevance in the mother-daughter experiments. Therefore, I find it rather confusing than helpful to include it as another dotted line in the Fig. 5 plots.

**Response:** We omitted "effusion" from the text and Fig. 5"

p.15/16 l. 365 - 394: The mass spectrometric analysis of isotope or molecule pair ratios appears to be a powerful tool to ascertain the fractionation process. Yet, as the presentation and discussion of the results currently are and with the conclusion drawn the reader feels left behind without much of a clue if the data confirm any assumption or not.
**Response:** We reformulated the parts (P16, L392-P17, L401).

p.15/16 l. 376 - 377: It is surprising that only for the two cases where the relationships for thermal diffusion from Langenfelds and Ishidoya investigation are in agreement the latter are not included in Fig. 5. This solid red line should be either added or it should be explained why it is not. The slopes of the other delta relationships in Figure 5 are varying quite a bit between the two references. The authors attribute this to "some factors that were not considered in the theoretical model " (= dotted line from Langenfelds 2005). However, the spread of the measured data presented here is of similar magnitude than this disagreement. This might therefore also point to an uncertainty in the accuracy of the experimentally determined literature value. Unfortunately, this discrepancy is particularly large for the D CO2/N2 vs D29N2/28N2 and the authors acknowledge: "Here, note that the deviation of the experimental thermal diffusion for $\delta$ (CO2/N2) has large uncertainty and requires further experiments.". It would help the reader if that statement could be either more quantitative or be explained more in detail. Is the underlying cause for that stated uncertainty an issue that might also apply to the measurements made in this study? Could information be provided on the measurement precision for the displayed delta-values (error bars on the black dots).
**Response:** We included all thermal diffusion data measured by Ishidoya in Fig.5. We also reformulated the parts (P16, L392-P17,L401).

p.15 l. 380: The following statement does not convince me: "The fact that the deviations of $\delta$ (CO2/N2) are close to the experimental thermal diffusion, indicates that the fractionations occurred by thermal diffusion." The agreement of the delta relationship is closer to the value from the Ishidoya studies than to the quoted Langenfelds number. However, on the one hand, their had just been doubts raised in l. 374 on that Ishidoya number, and on the other hand the observed deviation of the $\delta$ CO2/N2 ratio is also clearly larger than in the Ishidoya et al study.

In my opinion it does not help to vaguely point to "..unknown fractionation mechanisms.." or refer to diffusive processes occuring in firm that do not seem of any relevance to processes occuring in the mother daughter experiment with enormous bulk flow rates and without very small orifices.
**Response:** We reformulated the parts (P16, L392-P17, L401).

p.18 l. 437: The following sentence is incomprehensible: " These contributions are negligible to the increase because all cylinders act similarly.. ". What does it refer to (which contributions)?

".., although the fractionation in the transfer of the 2nd gas mixture into the daughter cylinder and ... also affect $CO_2$ molar fraction in the 3rd gas mixture." Why although? This is what has been explained to be the goal: to backtrack the impact of fractionation on the 2nd mixture by analysing the 3rd mixture.

I do not see the need for the sentence, but if I miss the message it should be reformulated.

**Response:** We revised the sentence to "The fractionation in the transfer of the 2nd gas mixture into the daughter cylinder and adsorption of $CO_2$ to the internal cylinder surface did not contribute to the increase of $CO_2$ molar fraction in the 3rd gas mixture because the effects act on all cylinders similarly." (P18, L444- P19,L447).

p.18 l. 445: The following sentence is confusing "The deviations increased by $0.25 \pm 0.10$ $\mu$ mol mol$-1$..." in that respect, that $CO_2$ increased but not the absolute value of the offset from the gravimetric value: at 11.5 MPa there is a deviation whereas at 2 MPa there is no deviation. You could either set the deviation from the gravimetric value of the 3rd mixture produced at 11.5 MPa pressure as reference (y-axis = 0), or re-phrase the sentence as e.g. "The known negative offset from the gravimetric value caused by the fractionation process in the gas transfer during the 3rd gas mixture production is observed for the 3rd prepared from the 2nd at 11.5 MPa. With decreasing 2nd pressure to 1.1 MPa $CO_2$ increased in the 3rd gas mixture by $0.25 \pm 0.10$ $\mu$ mol mol$-1$"

**Response:** We revised the sentence according to your comment (P19, L452-455).

p.18 l. 447ff: I am not sure if I understand what has been made here : "..we estimated the fractionation factor..": have the six data points shown in Fig. 8a and Eq. 4 been used to derive a new fractionation factor by this? Should be re-phrased.

**Response:** We revised the sentence to "we estimated the fractionation factor in the third step dilution by applying the Rayleigh fractionation model [the Eq. (4)] to the increase of the $CO_2$ mole fraction with the decrease of inner pressure," (P19, L457-459).

p.20 l. 482-482: Why is this described here again? Is this different from the calculation of measured values made before?

**Response:** This sentence is our mistake and this calculation is same as that made before. We therefore removed the sentence.

p.20 l. 490-492: I would remove the two concluding sentences here. They are well placed in the next section.

**Response:** We moved the two concluding sentences to conclusion.

p.21 l. 504: The data in Fig. 4c do not show that the decrease in CO2 is " weakened significantly with decreasing of the transfer speed". In this data set the decrease is independent from the flow rate in the range from 19 - 216 L/min! Needs to be re-phrased, e.g. " The decrease in CO2 molar fractions in the daughter cylinders does not depend on the transfer volume and initial pressure. This fractionation effect is neither depending on the transfer speed at flow rates exceeding 19 L/min but significantly reduced at lower flow rates ...

**Response:** We reformulated these sentences (P21, L509-515).

p.21 l. 513f: I suggest to rephrase: "The fractionation caused the CO2 molar fraction to increase and decrease. The reproducibility of CO2 molar fractions in gravimetric standard mixtures will suffer as a result." to "Fractionation at different stages of a multi-step dilution can result in CO2 increases as well as in CO2 decreases of the final gas mixture. This affects the reproducibility and accuracy of CO2 molar fractions in gravimetric standard gases"

**Response:** We revised the sentence according to your comment (P21,L518-P22, L523).

p.32 Fig. 5 top panel, y-axis: CO2/28N2)(permeg)

**Response:** We calculated CO2/N2 from equation 1 by substituting O2/N2 and Ar/N2 estimated from the values measured by the mass spectrometer.

Language

p.1 l. 16.: ..only once. It indicates...

**Response:** We revised the sentence according to your comment (P1, L12).

p.1 l.25: Needs to be re-phrased to become clear, as the following wording is probably wrong: ""The fact that the CO2 molar fraction weakened significantly...". Is the intention to state that the CO2 mole fraction (in the escaping?) gas was reduced when the transfer speed was

reduced? I presume it means to say: "The fact that the CO2 fractionation effect was less significant as the transfer speed decreased.."

**Response:** We revised the sentence to "The fact that the CO2 fractionation effect was less significant when the transfer speed decreased less than 3 L min$-1$ " (P1, L23-24).

p.1 l.27: The following would be easier to comprehend: "Experiments were conducted where a CO2 in air mixture was emitted from a cylinder to evaluate the CO2 adsorption..."

**Response:** We revised the sentence according to your comment (P1, L25).

p.2 l.29: rephrase, e.g. "...the CO2 molar fraction in the exiting gas mixture increased by 0.16 ± 0.04 μmol mol-1."

**Response:** We revised the sentence according to your comment (P2, L27- L28).

p.2 l. 39: It would be appropriate to provide references for the primary standard laboratories

**Response:** We revised the sentence according to your comment (P2, L36).

P.2 l45: ..that CO2 adsorbed on the internal...

**Response:** We revised the sentence according to your comment (P2, L44).

p.3 l. 65: Langenfelds...

**Response:** We revised the sentence according to your comment (P3, L64).

p.3 l.76f: Rephrasing the sentence would improve its clarity, e.g. : "..daughter cylinder causing an increase in the CO2 molar fraction in the remaining source gas in the mother cylinder".

**Response:** We revised the sentence according to your comment (P3, L73-76).

p.3 l. 77: Rephrasing the sentence would improve its clarity, e.g. : "This could be a factor that deteriorates the reproducibility of the assigned CO2 molar fractions because CO2 molar fractions in the prepared standard mixtures are biased by the decrease and increase in CO2 in the transfered gas mixture and the remaining pre-mixture, respectively."

**Response:** We revised the sentence according to your comment (P3, L76-P4,L78).

p.4 l. 88-90: The need of the sentence is not clear to me. I do not understand what the "Although.." aims to qualify.

**Response:** The sentence explains the reason to perform the experiments using 10 L aluminum cylinders. We revised the sentence alittle. (P4, L88-P4,L90).

p.4 l. 99: ".. estimate $CO_2$ adsorption on the internal..."
**Response:** We revised the sentence according to your comment (P4, L99).

p.5 l. 13-15: "The Picarro G2301 output was linearly calibrated using one standard mixture containing atmospheric $CO_2$ levels with a standard uncertainty of less than 0.1 $\mu$ mol mol$-1$."
**Response:** We revised the sentence according to your comment (P5,L113-115).

p.12 l. 280: "The fractionation of $CO_2$ and air during the transfer of a gas mixture with atmospheric $CO_2$ level has can be caused not only by the diffusive process but also the adsorption process."
**Response:** We revised the sentence according to your comment (P12,L300-302).

p.12 l. 289: "Therefore, the fractionation of $CO_2$ and air is expected to result from the diffusive fractionation process based on the three types of diffusion,."
**Response:** We revised the sentence according to your comment (P12,L292-293)..

13 l. 322: change "However,.." to "Correspondingly,.."
**Response:** We revised the sentence according to your comment(P15,L342).

14 l. 340: ".. of standard mixtures with accurate atmospheric $CO_2$..."
**Response:** We revised the sentence according to your comment (P13,L356).

p.21 l. 510: "We demonstrated that $CO_2$ molar fractions in standard mixtures by three-step dilutions decreased by $-0.207 \pm 0.060$ $\mu$ mol mol$-1$ from gravimetric values based on source gas fractionation, which is greater than the compatibility goal of 0.1 $\mu$ mol mol$-1$."
**Response:** We revised the sentence according to your comment (P21,L512-515).

---

## Author Response (AR2)

Response to reviewer 1

**General:**

The manuscript presents new updated CO2 concentration (and auxiliary) measurements regarding fractionations associated with adsorption/desorption on metal surfaces as well as fractionations during decanting experiments (mother-daughter, dilution experiments). These fractionations, in particular the latter, are relevant for assigning concentration and isotope values as best as possible as it significantly changes the concentration values. Aoki et al., did a thorough experimental study combined with explanations using models that has been used earlier on. Their results clearly show that care must be taken during dilution and decanting (mother-daughter) experiments.

This manuscript deserves publication in AMT after the manuscript has been checked for English language shortcomings as mainly addressed by Reviewer 1 and additionally outlined below.

Response: We appreciate the reviewer's thoughtful review and constructive comments. All the comments have been addressed in the revised manuscript, and the responses to your comments are given below.

**Minor points:**

Title: consider changing to: Influence of CO2 adsorption on cylinders and the fractionation of CO2 and air during the production of a standard mixture

**Response:** we revised title according to your comment.

Abstract: We conducted a study to fully understand carbon dioxide (CO2) adsorption
I do not know whether you should write it in such an absolute manner, consider skipping fully or exchange it with better.

**Response:** we revised the sentence to "We evaluated carbon dioxide (CO2) adsorption". (P1, L11)

Abstract: The CO2 molar fractions in standard mixtures prepared by diluting pure CO2 with air three times deviated by $-0.207 \pm 0.060$ $\mu$ mol mol$-1$ on average from the gravimetric values which were calculated from masses of source materials by evaluating their CO2 molar fractions based on standard mixtures by diluting the pure CO2 with the air only once.
This sentence is difficult to understand, consider splitting it up.

**Response:** We revised the sentence to "The CO2 molar fractions in the standard mixtures deviated from the gravimetric values by $-0.207 \pm 0.060$ $\mu$ mol mol$-1$ on average, larger

than the compatibility goal (0.1 $\mu$mol mol$-1$) recommended by the World Meteorological Organization." (P1, L13-15)

Abstract: rewrite: When the cylinder pressure was reduced from 11.0 to 0.1 MPa, the CO2 mole fractions in the mixture stream exiting the cylinder increased by 0.16 ± 0.04 $\mu$mol mol-1.

**Response:** We revised the sentence according to your comment. (P1, L24 - L26)

Intro: However, the compatibility goal has not been achieved among laboratories using their scales (Tsuboi et al., 2017, Flores et al., 2019), preventing precise evaluation of sources and sinks of CO2.

Here, I agree with Reviewer 1. Your conclusion is indeed misleading as the accuracy within the WMO GAW network does not play role as all the values needs to be reported on the same scale. The accuracy of the scale itself is of secondorder. Your cited references document differences among different scales in use.

Please reformulate this part.

**Response:** we reformulated this part according to your comment. (P2. L34-40)

Line 105-106: The mixture flow after through the regulator was branched to two ways by Tpieces. T, rephrase to "After flowing through the regulator, the mixture flow was branched in two ways by T-pieces.

**Response:** We revised the sentence according to your comment. (P55, L100-101)

Line 141: ⋯.(N2+O2,+Ar+CO2=1). Use ⋯.(N2+O2+Ar+CO2=1)

**Response:** We revised the sentence according to your comment. (P6, L143)

Line 507-510:The fractionation factor in the transfer of the CO2/Air mixture was 0.99968 ± 0.00010, indicating that the CO2 molar fraction decreased by 0.032 % ± 0.010 % by transfer of a source gas and the CO2 molar fraction in a source gas increases by 0.30 ± 0.10 $\mu$mol mol$-1$ as the inner pressure decreased from 11.5 MPa to 1.1 MPa.

rephrase this sentence.

**Response:** We reconstructed the sentences. (P21, L506-511)

Fig. 4: refer to subgraphs a), b) and c) in the Figure legend

**Response:** we revised Fig.4 according to your comment.

Response to reviewer 2

RC1: 'Comment on amt-2022-41', Anonymous Referee #2, 11 May 2022 reply
The manuscript describes experimental work to quantify effects that affect the mole fraction of $CO_2$ in standard air mixtures prepared by gravimetric methods. This certainly is a highly relevant topic within the scope of AMT. The topic has been addressed already by other papers that are adequately referenced. The experimental work has been performed thoroughly, the results of many individual experiments are very consistent and such add valuable quantitative data on the adsorptive loss of $CO_2$ on high pressure cylinder surfaces and fractionation effects when transfering gas mixtures from one cylinder to another for the purpose of preparing diluted samples. In previous studies of the investigation of such adsorptive effects results it was not always unambiguous if those results might also have been impacted by thermal fractionation. The very good agreement of several decanting experiments performed in this manuscript at different low flow rates yielding similar adsorbed quantities of $CO_2$ at cylinder walls as others have also published provides further confirmation of the quantitative relevance of this effect.

The results of the second series of experiments on fractionation during gas mixture transfer from one cylinder to an evacuated second cylinder (mother-daughter and dilution experiments) show clearly that the $CO_2$ mole fraction is altered by the process. It is shown that this effect is quantitatively very reproduceable with the experimental setup used for these experiments. This setup normally involves high flow rates for the gas transfer that are probably the cause for the disturbance. Three mother-daughter experiments have been conducted at lower transfer speeds resulting in a lower fractionation. However, this mode of operation has not been further employed at the panel utilized apparently is not made for such low flows. At this point the description of the experimental setup has not been sufficiently detailed to understand this. Isotope ratio mass spectrometry has been used to study the relationships in fractionation of various isotope or molecule pairs (e.g. $29N_2/28N_2$; $34O_2/32O_2$; $Ar/N_2$; $O_2/N_2$;..) and compare these measured relationships with literature results as an indicator for the kind of diffusive process causing the fractionation. These measurements apparently are not trivial and neither could a convincing conclusion be drawn from these data nor appears the discussion of these results very substantial.

There is a problem with the language being not sufficiently clear in several sentences throughout the manuscript that requires a thorough revision. For some key sentences I have provided suggestions below but this is certainly not exhaustive.

Concrete points that I feel the authors need to further elaborate or correct before final publication are listed here below.

**Response:** We appreciate the reviewer's thoughtful review and constructive comments. All the comments have been addressed in the revised manuscript, and the responses to your comments are given below.

lines 40-44: The conclusion drawn here is misleading and therefore should not remain. The quoted WMO compatibility goal is part of WMO recommendations that request to refer to the WMO CO2 mole fraction scale. By using this approach the absolute accuracy of the scale is of less importance and not a major contribution to the limit of the WMO GAW network compatibility. However, the two comparison studies that are refered to (Tsuboi et al, 2017 and Flores et al 2019) both comprise participating laboratories that mostly report on different scales. In the Japanese study it appears that for each lab the consistency of offsets of three comparison samples relative to NIES is very good for all the labs and the agreement between AIST and TU (both reporting on the TU2010 scale) is within the WMO compatibility goal. The comparison results of the CCQM Key Comparison (Flores et al 2019) provides the result of measurements performed at the BIPM of standard gases provided by the participating NMIs and the WMO Central Calibration Laboratory. Both references therefore do not provide any information on the compatibility of measurements by different laboratories relating to the WMO scale that might limit an evaluation of the carbon cycle.

**Response:** we reformulated this part according to your comment (P2, L36-40).

l.81: please insert reference: ".. into another cylinder (Hall et al. 2019)"

**Response:** We inserted the reference according to your comment(P4, L75).

p.5 l. 120: " some of these mixtures were purchased from a gas supplier (Japan Fine Products, Japan), while others were prepared at our laboratory.". This does not state if the gas mixtures are synthetic (which becomes clear later in the manuscript) or based on real ambient air. This should be stated explicitly here. What is confusing is the statement made at a later point (p. 6 l. 143) that "Atmospheric values of N2, O2, Ar ... were used as values of N2, O2, Ar.". This does not appear valid for synthetic mixtures.

**Response:** We stated information of the gas mixtures (P5, L115-116).

p.5/6 l. 127f: I guess that I do not understand the following sentence and it is not in clear agreement with Fig 1a: "The transfer speed was controlled using the only diaphragm valve

with the daughter cylinders calculated roughly from the transfer time and volume."

It appears that the full pressure of the mother cylinder is applied to the system with the evacuated daughter cylinder at the other end. This should cause a dramatically large flow entailing big temperature changes in the Mother and Daughter cylinders. There are several valves depicted in the figure that the gas is passing through. I do not know the type of diaphragm valve so what restriction it does provide. The one pressure gauge in Fig. 1a is downstream of diaphragm valves from the Mother cylinders and upstream of the diaphragm valve of the Daughter cylinders so it is not clear to me how it is checked when the transfer volume has been reached. How is the transfer time then measured? How long is the transfer typically taking? In Fig. 1a there are two cylinders named Mother and one Daughter. Please revise to make it clear.

**Response:** These parts had not been clear. We explained the transfer method of the mixtures in detail by revision of sentences and Figure 1a (P5, L126 - P6, L131).

p 6. l 141: It would be helpful if the equation 1 could be explicitly derived.

**Response:** We revised the equation 1 and the sentence about the equation 1(P6, L144).

p 6. l. 149: Could you please specify what is G1-grade air?

**Response:** We revised the sentence to "purified air (G1-grade ($< 0.1$ $\mu$mol mol$-1$ for CO, CO2, THC, $< 0.01$ $\mu$mol mol$-1$ for NOx, SO2, $< -80$ ℃ for H2O), Japan Fine Products, Japan)" (P7, L150-152).

p.7 l. 158: Was there any significant humidity detectable in the air?

**Response:** H2O concentration in the air was under 0.1 ppm.

p.8 l. 184ff: It is not explained why the 0.8L cylinder is moved from V6 to V10. Is it in order to avoid any CO2 getting lost in the the valves V6-V9? If so, how has been assured that no additional CO2 that might have adsorbed in valves V1 - V7 is later flushed into the 10 L cylinder during the 300 pressure expansion cycles with the dilution air?

**Response:**  We revised Figure1b and sentences (P8, L183-188). We had confirmed that additional CO2 that have adsorbed in valves was not detected by measuring Air flushed in the manifold after pure CO2 was transferred although we did not describe it in the paper.

p.8 l. 197: Was this preparation done using the set-up displayed in Fig. 1a?

**Response:** It was done using the set-up displayed in Fig. 1a. We added "by using the manifold

shown in Fig 1c." at the end of the sentence (P8, L200).

p.9 l. 215f: "The output of the Picarro G2301 was calibrated using standard mixtures prepared by the one-step dilution." In section 2.2.3 it is not specified how many one-step dilution standards are prepared in parallel. This information is provided at a later stage but number of calibration points and the CO2 mole fraction range covered should also be mentioned here.
**Response:** We added the information after "The output of the Picarro G2301 was calibrated using standard mixtures prepared by the one-step dilution." (P10, L221-234).

p.9 l. 223: "However, diffusive fractionation in the transfer of the source gas is likely to have a larger impact on the CO2 molar fractions than the bias by the sorption processes." This appears like an anticipation of the results the study is concluding with. It also relates to the large flow rates employed in this experimental set-up.
**Response:** we revised "Furthermore, the transfer of the CO2/air mixture changed CO2 molar fractions by about 0.10 $\mu$ mol mol$-1$. The transfer of source gases impacts the CO2 molar fractions more strongly compared to the deviation on the adsorption process. " (P10, L228-230)

p.11 l. 250: Omit the first sentence ("The standard deviation is..")
**Response:** We omitted the sentence (P11, L252).

p.12 l. 297-299: This sentence needs to be corrected as follows: " Lighter molecules preferentially escape from the orifice relative to heavier ones because the rate of effusion is inversely proportional to the square root of the mass".
**Response:** We removed the sentence because sentences about effusion were reformulated (P11, L292- P12,301).

p.13 l. 306: Is the diffusive process quick enough to cause a fractionation because of the pressure gradient between Mother and Daughter cylinder entail very high bulk flow rates?
**Response:** We think that the fractionation occurs in the mother cylinders. Mother-daughter experiments from horizontal mother cylinders into the daughter cylinders suggested that thermal diffusion was main factor of the fractionation. Therefore, we added the experiment results in Fig.4 and revised the text (P13, L308- L319).

p.13 l. 307f: I would think this sentence: "Additionally,.." should be omitted, as all these diffusive processes are based on molecular mass differences.

**Response:** We omitted the sentence according to your comment.

p.13 l. 313f / Fig. 4 / Fig. 1b: The description of the panel used for the mother-daughter experiments depicted in Fig. 1b does not make clear how the transfer speed is controlled but Fig. 4 clearly indicates that in the experimental series transfer speed and transfer volume were independently controlled. Therefore, please do clarify in Fig. 1a or section 2.1 by what means (which type of valves) the flow can be regulated.
**Response:** We revised Fig. 1a and explained how the transfer speed is controlled in the section 2.1 (P6, L123- P7, L131).

p.14 l. 333: The statement "..,but it becomes significantly weaker as the transfer speed decreases" suggests a continuous relationship which the data do not show. Rather change to "..but it becomes significantly smaller below flow rates of 19 L/min"
**Response:** We revised the sentence according to your comment (P14, L345).

p.14 l. 336: Why would you consider Knudsen diffusion to play a role? The orifice of the valves certainly will exceed the free path length of molecular collisions. Omit "effusion".
**Response:** As you describe, the orifice will exceed the free path length of molecular collisions. We omitted "effusion".

p.14 l. 344: This sentence poses again the question: how has the flow been regulated to perform the experiments resulting in the transfer speeds < 3L/min (last three data points of Table 1 and open triangles in Fig. 4) ?
**Response:** We explained how the transfer speed is controlled in the section 2.1.

p.15 l. 333: Does the statement "..using the values in Table 1." include alle the values (that is what is written) or only values with transfer speeds ≥ 19 L/min (that would make more sense to me, as apparently the fractionation factor is not constant below some threshold between 3 and 19 L/min)?
**Response:** We thought that the statement "…using the values in Table 1" include only values with transfer speeds 19 L/min. We revised the sentence to "…using only values with transfer speeds 19 L/min in Table 1" (P15, L362)

p.15 l. 372: As stated above, Knudsen diffusion ("effusion") will not be of relevance in the mother-daughter experiments. Therefore, I find it rather confusing than helpful to include it as another dotted line in the Fig. 5 plots.

**Response:** We omitted "effusion" from the text and Fig. 5"

p.15/16 l. 365 - 394: The mass spectrometric analysis of isotope or molecule pair ratios appears to be a powerful tool to ascertain the fractionation process. Yet, as the presentation and discussion of the results currently are and with the conclusion drawn the reader feels left behind without much of a clue if the data confirm any assumption or not.
**Response:** We reformulated the parts (P16, L383- L391).

p.15/16 l. 376 - 377: It is surprising that only for the two cases where the relationships for thermal diffusion from Langenfelds and Ishidoya investigation are in agreement the latter are not included in Fig. 5. This solid red line should be either added or it should be explained why it is not. The slopes of the other delta relationships in Figure 5 are varying quite a bit between the two references. The authors attribute this to "some factors that were not considered in the theoretical model " (= dotted line from Langenfelds 2005). However, the spread of the measured data presented here is of similar magnitude than this disagreement. This might therefore also point to an uncertainty in the accuracy of the experimentally determined literature value. Unfortunately, this discrepancy is particularly large for the D CO2/N2 vs D29N2/28N2 and the authors acknowledge: "Here, note that the deviation of the experimental thermal diffusion for $\delta$ (CO2/N2) has large uncertainty and requires further experiments.". It would help the reader if that statement could be either more quantitative or be explained more in detail. Is the underlying cause for that stated uncertainty an issue that might also apply to the measurements made in this study? Could information be provided on the measurement precision for the displayed delta-values (error bars on the black dots).
**Response:** We included all thermal diffusion data measured by Ishidoya in Fig.5. We also reformulated the parts (P16, L383- L391).

p.15 l. 380: The following statement does not convince me: "The fact that the deviations of $\delta$ (CO2/N2) are close to the experimental thermal diffusion, indicates that the fractionations occurred by thermal diffusion." The agreement of the delta relationship is closer to the value from the Ishidoya studies than to the quoted Langenfelds number. However, on the one hand, their had just been doubts raised in l. 374 on that Ishidoya number, and on the other hand the observed deviation of the $\delta$ CO2/N2 ratio is also clearly larger than in the Ishidoya et al study.

In my opinion it does not help to vaguely point to "..unknown fractionation mechanisms.." or refer to diffusive processes occuring in firn that do not seem of any relevance to processes occuring in the mother daughter experiment with enormous bulk flow rates and without very

small orifices.

**Response:** We reformulated the parts (P16, L383- L391).

p.18 l. 437: The following sentence is incomprehensible: " These contributions are negligible to the increase because all cylinders act similarly.. ". What does it refer to (which contributions)?

".., although the fractionation in the transfer of the 2nd gas mixture into the daughter cylinder and ... also affect CO2 molar fraction in the 3rd gas mixture." Why although? This is what has been explained to be the goal: to backtrack the impact of fractionation on the 2nd mixture by analysing the 3rd mixture.

I do not see the need for the sentence, but if I miss the message it should be reformulated.
**Response:** We revised the sentence to "he decrease amounts of the CO2 molar fractions in the 2nd gas mixture transferred into the daughter cylinder is same for all 3rd gas mixtures, because the effects on the transferred mixtures act similarly." (P18, L433- L435).

p.18 l. 445: The following sentence is confusing "The deviations increased by $0.25 \pm 0.10$ $\mu$ mol mol$-1$..." in that respect, that CO2 increased but not the absolute value of the offset from the gravimetric value: at 11.5 MPa there is a deviation whereas at 2 MPa there is no deviation. You could either set the deviation from the gravimetric value of the 3rd mixture produced at 11.5 MPa pressure as reference (y-axis = 0), or re-phrase the sentence as e.g. "The known negative offset from the gravimetric value caused by the fractionation process in the gas transfer during the 3rd gas mixture production is observed for the 3rd prepared from the 2nd at 11.5 MPa. With decreasing 2nd pressure to 1.1 MPa CO2 increased in the 3rd gas mixture by $0.25 \pm 0.10$ $\mu$ mol mol$-1$"
**Response:** We revised the sentence according to your comment (P18, L440-443).

p.18 l. 447ff: I am not sure if I understand what has been made here : "..we estimated the fractionation factor..": have the six data points shown in Fig. 8a and Eq. 4 been used to derive a new fractionation factor by this? Should be re-phrased.
**Response:** We revised the sentence to "we estimated the fractionation factor in the third dilution step by applying the Rayleigh fractionation model [Eq. (4)] to the increase in the CO2 mole fraction with the decrease in inner pressure," (P19, L445-447).

p.20 l. 482-482: Why is this described here again? Is this different from the calculation of

measured values made before?

**Response:** This sentence is our mistake and this calculation is same as that made before. We therefore removed the sentence.

p.20 l. 490-492: I would remove the two concluding sentences here. They are well placed in the next section.

**Response:** We moved the two concluding sentences to conclusion.

p.21 l. 504: The data in Fig. 4c do not show that the decrease in CO2 is " weakened significantly with decreasing of the transfer speed". In this data set the decrease is independent from the flow rate in the range from 19 - 216 L/min! Needs to be re-phrased, e.g. " The decrease in CO2 molar fractions in the daughter cylinders does not depend on the transfer volume and initial pressure. This fractionation effect is neither depending on the transfer speed at flow rates exceeding 19 L/min but significantly reduced at lower flow rates ...

**Response:** We reformulated these sentences (P20, L497-501).

p.21 l. 513f: I suggest to rephrase: "The fractionation caused the CO2 molar fraction to increase and decrease. The reproducibility of CO2 molar fractions in gravimetric standard mixtures will suffer as a result." to "Fractionation at different stages of a multi-step dilution can result in CO2 increases as well as in CO2 decreases of the final gas mixture. This affects the reproducibility and accuracy of CO2 molar fractions in gravimetric standard gases"

**Response:** We revised the sentence according to your comment (P21,L506- L508).

p.32 Fig. 5 top panel, y-axis: CO2/28N2)(permeg)

**Response:** We calculated CO2/N2 from equation 1 by substituting O2/N2 and Ar/N2 estimated from the values measured by the mass spectrometer.

Language

p.1 l. 16.: ..only once. It indicates...

**Response:** We revised the sentence (P1, L13-L15).

p.1 l.25: Needs to be re-phrased to become clear, as the following wording is probably wrong: ""The fact that the CO2 molar fraction weakened significantly...". Is the intention to state that

the CO2 mole fraction (in the escaping?) gas was reduced when the transfer speed was reduced? I presume it means to say: "The fact that the CO2 fractionation effect was less significant as the transfer speed decreased.."

**Response:** We revised the sentence to "The CO2 fractionation was less significant when the transfer speed decreased to less than 3 L min$-1$ " (P1, L21-22).

p.1 l.27: The following would be easier to comprehend: "Experiments were conducted where a CO2 in air mixture was emitted from a cylinder to evaluate the CO2 adsorption..."

**Response:** We revised the sentence to "The CO2 adsorption on the internal cylinder surface was experimentally evaluated by emitting a CO2/air mixture from a cylinder." (P1, L23-L24).

p.2 l.29: rephrase, e.g. "...the CO2 molar fraction in the exiting gas mixture increased by 0.16 $\pm$ 0.04 µmol mol-1."

**Response:** We revised the sentence according to your comment (P2, L25- L26).

p.2 l. 39: It would be appropriate to provide references for the primary standard laboratories

**Response:** We revised the sentence according to your comment (P2, L33-L40).

P.2 l45: ..that CO2 adsorbed on the internal...

**Response:** We revised the sentence according to your comment (P2, L41).

p.3 l. 65: Langenfelds...

**Response:** We revised the sentence according to your comment (P3, L60).

p.3 l.76f: Rephrasing the sentence would improve its clarity, e.g. : "..daughter cylinder causing an increase in the CO2 molar fraction in the remaining source gas in the mother cylinder".

**Response:** We revised the sentence according to your comment (P3, L70-71).

p.3 l. 77: Rephrasing the sentence would improve its clarity, e.g. : "This could be a factor that deteriorates the reproducibility of the assigned CO2 molar fractions because CO2 molar fractions in the prepared standard mixtures are biased by the decrease and increase in CO2 in the transfered gas mixture and the remaining pre-mixture, respectively."

**Response:** We revised the sentence according to your comment (P3, L71-L73).

p.4 l. 88-90: The need of the sentence is not clear to me. I do not understand what the "Although.." aims to qualify.

**Response:** The sentence explains the reason to perform the experiments using 10 L aluminum cylinders. We revised the sentence a little. (P4, L81- L83).

p.4 l. 99: ".. estimate CO2 adsorption on the internal..."
**Response:** We revised the sentence according to your comment (P4, L85).

p.5 l. 13-15: "The Picarro G2301 output was linearly calibrated using one standard mixture containing atmospheric CO2 levels with a standard uncertainty of less than 0.1 $\mu$ mol mol$-1$."
**Response:** We revised the sentence according to your comment (P5,L106-108).

p.12 l. 280: "The fractionation of CO2 and air during the transfer of a gas mixture with atmospheric CO2 level has can be caused not only by the diffusive process but also the adsorption process."
**Response:** We revised the sentence according to your comment (P12,L294-296).

p.12 l. 289: "Therefore, the fractionation of CO2 and air is expected to result from the diffusive fractionation process based on the three types of diffusion,."
**Response:** We revised the sentence according to your comment (P12,L285-286)..

13 l. 322: change "However,.." to "Correspondingly,.."
**Response:** We revised the sentence according to your comment(P15,L334).

14 l. 340: ".. of standard mixtures with accurate atmospheric CO2..."
**Response:** We revised the sentence according to your comment (P14,L350).

p.21 l. 510: "We demonstrated that CO2 molar fractions in standard mixtures by three-step dilutions decreased by $-0.207 \pm 0.060$ $\mu$ mol mol$-1$ from gravimetric values based on source gas fractionation, which is greater than the compatibility goal of 0.1 $\mu$ mol mol$-1$."
**Response:** We revised the sentence according to your comment (P21,L501-503).